# Spin Density Topology

**DOI:** 10.3390/molecules25153537

**Published:** 2020-08-02

**Authors:** Giovanna Bruno, Giovanni Macetti, Leonardo Lo Presti, Carlo Gatti

**Affiliations:** 1Dipartimento di Chimica, Università degli Studi di Milano, via Golgi 19, 20133 Milano, Italy; giovanna.bruno@unimi.it (G.B.); leonardo.lopresti@unimi.it (L.L.P.); 2Laboratoire de Physique et Chimie Théoriques (LPCT), Université de Lorraine & CNRS, 1 Boulevard Arago, F–57078 Metz, France; giovanni.macetti@univ-lorraine.fr; 3CNR–SCITEC, Istituto di Scienze e Tecnologie Chimiche sezione di via Golgi, c/o Università degli Studi di Milano, via Golgi 19, 20133 Milano, Italy; 4Istituto Lombardo, Accademia di Scienze e Lettere, via Brera 28, 20100 Milano, Italy

**Keywords:** spin density, topology, quantum chemical topology, spin density critical points, spin maxima and minima joining paths, molecular spin graph, spin density basins, spin density source function, water ^3^B_1_ triplet

## Abstract

Despite its role in spin density functional theory and it being the basic observable for describing and understanding magnetic phenomena, few studies have appeared on the electron spin density subtleties thus far. A systematic full topological analysis of this function is lacking, seemingly in contrast to the blossoming in the last 20 years of many studies on the topological features of other scalar fields of chemical interest. We aim to fill this gap by unveiling the kind of information hidden in the spin density distribution that only its topology can disclose. The significance of the spin density critical points, the 18 different ways in which they can be realized and the peculiar topological constraints on their number and kind, arising from the presence of positive and negative spin density regions, is addressed. The notion of molecular spin graphs, spin maxima (minima) joining paths, spin basins and of their *valence* is introduced. We show that two kinds of structures are associated with a spin–polarized molecule: the usual one, defined through the electron density gradient, and the *magnetic* structure, defined through the spin density gradient and composed in general by at least two independent spin graphs, related to spin density maxima and minima. Several descriptors, such as the spin polarization index, are introduced to characterize the properties of spin density critical points and basins. The study on the general features of the spin density topology is followed by the specific example of the water molecule in the ^3^B_1_ triplet state, using spin density distributions of increasing accuracy.

## 1. Introduction

### 1.1. Introductory Remarks and Scope

The electron density ρ is a sum of two components due to the α– and the β–spin electrons. When ρ_α_ and ρ_β_ are not equal, there is spin polarization and the spin density s, given by their difference, is not zero [1] (p. 35). The spin density s and the electron density ρ are the two 3D space functions acting as basic variables in spin density functional theory (SDFT), which is the required generalization of the DFT in the presence of a magnetic field, besides the usual scalar external potential due to the nuclei. Even when this additional potential is absent, SDFT is of importance as “it allows to build more physics into the approximate exchange–correlation functional through its spin–dependence” [1] (p. 169). As a result, a more correct treatment of spin–polarized systems, such as open–shell atoms and molecules or molecules near the dissociation limit, is possible [1]. 

Apart its function in SDFT, the spin density reflects the interactions that are taking place among the electron spins within a system and so also plays a fundamental role for understanding magnetic phenomena [2,3,4,5,6]. For example, the detailed knowledge of the spin density distribution in a molecule is essential for interpreting how spin polarization propagates in molecular complexes and crystals, allowing one to highlight and rationalize the various magnetic interactions as a function of molecular orientation and packing [4,7,8]. These interactions are comparable or even smaller in energy than the thermal energy, *k_B_T*, yet they can be exploited for designing magnetic materials with foreseeably important technological applications [3]. 

Despite the importance of the electron spin density, only a few studies have appeared on its subtleties thus far. We are not aware of a systematic topological analysis of this function, seemingly in contrast with the blossoming in the last 20 years of many topological analyses of scalar fields related to chemistry and defined in terms of quantum mechanical functions [9,10,11,12]. Among these studies, which are collectively gathered under the umbrella of *Quantum Chemical Topology* [9,10], a dominant place is undoubtedly taken by those on the electron density, the spin density companion key ingredient function in SDFT. Such a prominent position is justified by the cornerstone role that the electron density plays in the Bader’s Quantum Theory of Atoms in Molecules (QTAIM) [13]. In QTAIM, all elements of chemical structure—atoms, bonds, rings, cages and the structure itself—can be rigorously defined through the electron density topology. 

The aim of the present work is to unveil the information hidden in the spin density distribution that only its topology can disclose. Questions such as the meaning of the spin density critical points (where ∇s = 0), the physical conditions for their occurrence, the different ways they may realize depending on their kind (maxima, saddles, minima) and on the relative dilution or concentration of the spin majority and the spin minority distributions at the critical point, are first addressed. Then the notions of spin graphs, spin molecular structure, spin maxima and minima joining paths, spin basins and their *valence* are introduced. The study is organized into two intimately connected parts: the first one is strictly general and of a fundamental nature (Section 2), while the second one is related to the exemplary case of the water molecule in the ^3^B_1_ state and worked out in very much detail (Section 3). It is shown that a spin–polarized molecule, at variance with a non–polarized one, is described through two kinds of structures. One is customarily defined through the electron density gradient and is in a one–to–one correspondence with the empirical notion of chemical structure [13]. The other one, the *magnetic* structure, is obtained through the ∇s field and is in general composed by at least two independent spin graphs, associated with the spin density maxima and minima, respectively. While the conventional chemical structure and the spin density maxima magnetic structure are quite robust against changes in the way they are computed, the spin density minima magnetic structure is instead very much dependent on the way it is evaluated. This fact also has a clear impact on the number of recovered critical points and on the way the Poincaré–Hopf theorem [13,14,15,16], which expresses the topological constraint among the numbers of each kind of critical points, is separately satisfied in the positive and negative spin density regions. Properties of spin density basins, defined in two possible alternative ways, are discussed and some interesting indexes to characterize their spin polarization nature are proposed. The relative capability of the various spin basins to determine the spin density magnitude and sign in the various molecular regions is also explored, using the source function tool (see infra) [17,18,19]. The present study suggests that the spin density topology can disclose a wealth of interesting information. It so surely deserves further studies both on its fundamental aspects and on the exploration of its features in more complex systems. We are currently performing a few studies along these lines, using spin–polarized transition metal complexes as more difficult and, possibly, interesting cases. A next step will then involve the spin density topology of interacting complexes, like those studied in [4,5,6,7,8]. 

### 1.2. Literature Context

Before presenting our work on the spin density topology, we deem it important to briefly recall a few significant studies, which inspired us and introduce some terminology and useful concepts. All these studies were similarly aimed at disclosing important and general details of spin–polarized distributions as well as of the mechanisms of spin propagation from a paramagnetic center to the neighboring atoms or ligands or from a molecule to other molecules in a molecular complex or solid. In their 2005 review paper, Ruiz et al. [4] discussed the characteristic shapes of the spin density distributions around a transition metal atom, as a function of its electronic configuration. This analysis represents a first qualitative study of the topology of the spin density in metal coordination complexes. Ruiz et al. also summarized the characteristics trends involving the two main mechanisms through which an α– unpaired electron in a singly occupied molecular orbital (SOMO) can place some spin density at the other atoms of the molecule and at its own other atomic orbitals. One mechanism is *spin delocalization*. It is due to the contribution of the various atomic orbitals to the SOMO and results in a distribution of positive spin density throughout the molecule. The other one is *spin polarization*, through which the positive α spin at a center may induce some spin density of the opposite sign at the atoms bonded to it. Spin polarization is due to electron exchange, which reduces the electron repulsion between electrons of the same–spin and favors their proximity relative to electrons of opposite spin. As a result, the α–electron of a bonding pair of electrons will preferentially stay closer to the atom of the bonded pair of atoms with higher excess of positive spin, while a corresponding β–spin polarization will be induced on the remaining atom. Thus, spin polarization induces chain(s) of alternate α– and β–spin–polarized atoms, which propagate from paramagnetic center(s) [4]. Both the mechanisms are always in operation and may either cooperate or oppose each other in determining the spin density magnitude and sign at any point in the real space. This description of the spin transmission mechanisms strictly applies to UHF (Unrestricted Hartree Fock) wavefunctions, while it is known that static and dynamic electron correlation effects, not considered in the Hartree–Fock formalism, also influence spin polarization [7]. 

In a series of studies [20,21], Bochicchio and coworkers showed how to decompose the electron density of open–shell systems into contributions of paired and effectively unpaired nature and studied some aspects of their quite distinct topologies. Their results showed that unpaired and paired electron–density concentrations occur in mutually exclusive regions, the former out of the topological bonding regions while the latter inside those regions. While different from a direct study of the spin density topology as we propose in our work, the approach devised by Bochicchio et al. is clearly appealing from both a physical and chemical point of view. Also related to our work and yet not performed directly on the spin density is the study by Jones et al. [22] on the topology of the majority and minority spin density components compared to the total (spin–polarized) density topology in bcc and fcc iron. They found that, while the total density owns the same topology of the non–magnetic prototypical structures, the two component densities possess quite distinct features. For bcc and high–spin fcc iron, the spin–minority density topology recovers directional bond paths and deep minima, while the spin–majority density tends to fill these holes, reducing bond directionality. 

A first experimental study of spin majority and spin minority densities, including few of their topological features, was reported by Deutsch et al. [23] on a ferromagnetic end–to–end asymmetric di–azido copper complex. These authors could derive both the electron position density ρ(**r**) and its resolved α and β spin components using a spin–split version [24] of the original Hansen and Coppens multipolar model [25] able to refine the parameters of a multipolar model against both X–ray and polarized neutron diffraction (PND) data. Unlike nuclear and electron resonance techniques, such as NMR and EPR, the more demanding polarized neutron diffraction (PND) experiment has the key advantage to provide a direct mapping of the spin density in the periodic solid [6,7,23]. Deutsch et al. [23] found that the (X–ray/PND)–refined *ρ_α_* and *ρ_β_* distributions were comparable to those determined through ab–initio methods. Both approaches found a spherical spin up distribution in the vicinity of the copper nucleus, while the spin down distribution showed maxima in the bisecting direction of the ligands.

However, the accurate evaluation of the spin resolved distributions difference s(**r**) is much more demanding for both experiment and theory [19,26,27,28,29,30]. Indeed, a qualitative agreement between spin resolved distributions obtained with two different methods has been proven to be insufficient as it may still hide serious discrepancies between their associated spin densities [29]. The problem of spin density accuracy has been carefully addressed by Conradie and Gosh [26] and by M. Reiher and coworkers [27,28]. Systematic studies using several DFT exchange–correlation functionals demonstrated that DFT is in general unable to treat open–shell systems properly, leading to non–accurate electron spin densities [26,27,28]. Ab–initio electron correlation approaches or the density–matrix renormalization group (DMRG) methods are required to predict reliable spin densities [28]. Yet, they are often computationally too demanding, especially for large systems. The complete active space self–consistent field (CASSCF or CAS) method, using an appropriate number of correlated electrons and a large enough active orbital space, often represents a good compromise, as this method yields spin densities comparable to those of converged DMRG results that serve as an (ideal) reference [28].

Despite considerable efforts, interpretation of the electron spin density and of its model–dependent subtleties is a difficult task and still an open issue [29,30]. To this purpose, two of us (C.G. and L.L.P.) have recently extended [19] to the spin density distribution the concept of the Source Function (SF), introduced back in 1998 by Bader and Gatti for the electron density [17]. Within the SF approach the value of a scalar function at any point in the molecular or crystalline space is envisaged as caused by *local source* contributions from all other points in the space. Using a suitable and exhaustive 3D space partitioning, like that in atomic basins of the QTAIM, the value of the electron density or of the electron spin density at a point may be expressed in terms of a sum of atomic SF contributions [19,29].

Hence, analogously to the standard SF, the electron spin density SF provides a quantitative appreciation of the relative capability of the atoms in a system to determine the spin density at given system’s locations. Applied first to the water molecule in the ^3^B_1_ state [19], then on alkyl radicals [31] and finally on two, more complex, azido Cu(II) dinuclear complexes [29,30], the electron spin density SF has revealed itself as a precious investigative tool. Within a cause–effect view and without adopting any orbital model, but just making use of an observable, the SF details how spin information is spread out from paramagnetic to non–magnetic centers. It also depicts whether an atom or a group of atoms in a system favours or opposes the paramagnetic centers in determining the spin density value at a given point and whether it does so in a relevant or negligible way. Further precious insight on these mechanisms was recovered by decomposing each spin density value and its SF additive contributions into *magnetic* and *relaxation* parts [19,29,30,31], the former due to the fully unpaired α–electrons density and the latter to the remaining α– and β–electrons density (in practice, the magnetic contribution is obtained from the singly occupied natural orbitals and the relaxation contribution from the remaining natural orbitals). The sketched decomposition proved also very valuable for discussing the accuracy of spin density distributions as a function of the adopted theoretical level. While the magnetic contribution is already reliably calculated at the crude Restricted Open Hartree Fock (ROHF) approach, a careful treatment of electron correlation seems to be required for an accurate evaluation of the relaxation effect [19,29,30,31]. Unrestricted DFT was generally found to exaggerate electron spin delocalization from paramagnetic center(s) towards their bonded atoms and to overestimate spin polarization effects, relative to CAS [19,29,30].

Compared to the SF study of the spin density, our present work on the spin density topology represents a further attempt to investigate, in a direct way, such an important observable, without explicitly recurring to any orbital approach. The spin density distribution and its topological features provide an observable and real space fingerprint of the consequence of the hypothesized spin delocalization and spin polarization mechanisms.

## 2. Theoretical Details

### 2.1. Topology of the Electron Spin Density

The electron spin density s,
s(**r**) = ρ_α_ (**r**) − ρ_β_(**r**)(1)
where ρ_α_ and ρ_β_ are the electron densities of the α– and β–spin electrons, expresses the local spin polarization. As is customary in the field of Quantum Chemical Topology (QCT) [9,10], this scalar function can also be analyzed through the theory of the gradient dynamical systems [9,32,33,34] and using its well–known entities, concepts and related terminology (e.g., attractor, basin, gradient path/phase curve, separatrix, critical points, homeomorphism). Thanks to this theory, chemical systems may be partitioned in space–exhaustive disjoint pieces, often bearing chemical significance, and then characterised by dissecting a global system property into a sum of the individual properties of such pieces. In the following, we will examine which kind of peculiar information is conveyed by applying such procedures to the electron spin density s rather than to more usual scalar fields as, for instance, the total electron density ρ, with ρ = ρ_α_ + ρ_β_.

Before doing so, we briefly recall that a *dynamical system* is defined as a field of bound vectors ***X*** on a manifold *M,* where in any of its points of coordinates {*m*} the equations d***m***/d*t* = ***X***(***m***) determine a unique trajectory *h*(***m***) [32,33]. In formal analogy with a velocity field, the variable *t* may be taken as a time variable and by integrating the equations d***m***/d*t* = ***X***(***m***) one obtains trajectories that begin and terminate in the neighbourhood of points where ***X***(***m***) = **0**. For a given point *p*
*∈*
*M*, α(*p*) and ω(*p*) denote, respectively, the *t* → −∞ and the *t* → +∞ limit–sets of *p(t)* in *M*. A *gradient dynamical system* [32,33] is then a dynamical system for which the vector field ***X*** derives from a scalar function *V*, called the *potential function*, that is ***X*** = **∇***V*. The potential function carries the physical or chemical information. Note that the word *potential* does not carry here the usual meaning it has in mechanics, where the gradient field of the potential energy is the force field, rather than the velocity field.

#### 2.1.1. Electron Spin Density Critical Points

A gradient dynamical system [32,33] *V* (the electron spin density s in the present case), is characterized by two main kinds of points, the wandering ones **∇***V* ≠ 0 and the *critical points* (CPs), where **∇***V* = 0. Except for asymptotic behaviours, the α and ω limits are CPs. The *stable manifold* of a CP is the set of all those points for which the CP is an ω–limit, while the *unstable manifold* corresponds to the set of all those points for which it is an α–limit.

The CPs play a key role in QCT methods and have often a profound chemical significance. For instance, in the case of the electron density topology and depending on their kind (see *infra*), they have been associated wiith the elements of chemical structure, i.e., nuclei, bonds, rings and cages [13].

Here, we consider the physical implications occurring at any electron spin density critical point **r**_c_,
(2)$$\nabla s(r_{c})=\nabla [\rho_{\alpha }(r_{c})-\rho_{\beta }(r_{c})]=0\;\;\;\;i.e.,\;\nabla \rho_{\alpha }(r_{c}{)}_{\;}=\nabla \rho_{\beta }(r_{c})\;\forall r_{c}$$

The condition given in Equation (2) implies that the derivatives of the ρ_α_ and the ρ_β_ distributions at such points are equal in both magnitude and direction, so that the electron spin density CPs will clearly differ in number and location relative to those of the total density ρ_,_ characterized by the opposite condition for the derivatives direction, ∇ρ_α_(**r**_c_) = −∇ρ_β_(**r**_c_). For a spin not polarized system, the condition at the electron density CPs becomes more stringent, as ∇ρ_α_(**r**_c_) and ∇ρ_β_(**r**_c_) also need to both be equal to zero. This event could apparently be seen as a special case of Equation (2). However, this specific condition does not realize in practice for a spin–polarized system since ρ_α_ and ρ_β_ generally differ at any point **r** and so have their own CPs at different locations. 

The CPs of a gradient dynamical system are usually classified according to *a)* their *index I*, given by the number of positive eigenvalues of the Hessian matrix of *V* at **r**_c_ and denoted, for a given CP *p*, as *I*(***X***;*p*), and *b)* by the pair of integers (*r; s*), where *r* is the rank (number of non–zero eigenvalues) and *s* the signature (number of positive minus negative eigenvalues) of this same Hessian matrix. There are four kinds of rank–3 (non–degenerate) critical points in *R*^3^: (*i*) *attractors* of index 0, or (3, −3) CPs, corresponding to local maxima of *V* and being only ω–limits; (*ii*) *saddle points of index* 1 or (3, −1) CPs, minima of *V* in one direction and maxima of *V* in the surface of their stable manifolds; (*iii*) *saddle points of index* 2 or (3, +1) CPs, maxima of *V* in one direction and minima of *V* in the surface of their unstable manifolds; (*iv*) *repellers* of index 3 or (3, +3) CPs which are local minima of (*V* and only α–limits. Saddle points act as both α– and ω–limits. 

Table 1 collects the four kinds of non–degenerate CPs of the electron spin density, yet introducing for each s CP kind an additional classification by s CP *type* in terms of the −∇^2^ρ_α_ and −∇^2^ρ_β_ Laplacian values at the CP. In fact, while (3, −3) and (3, +3) CPs imply positive and negative −∇^2^s value at CP, respectively, and (3, −1) and (3, +1) CPs allow for any spin density Laplacian sign, each CP kind may be further characterized by a number of allowed sign combinations and possible constraints on the relative magnitude of the –∇^2^ρ_α_ and –∇^2^ρ_β_ Laplacian values at the CP. As shown in the Table 1, there are three different types for (3, −3) CPs, six types each for both (3, −1) and (3, +1) CPs and three more types for (3, +3) CPs, yielding a total of eighteen different CP types. For instance, at an electron spin density maximum (3, −3) CP, −∇^2^s will be necessarily positive, but −∇^2^ρ_α_ and −∇^2^ρ_β_ may be also both positive provided −∇^2^ρ_α_ > −∇^2^ρ_β_ (corresponding to a CP maximum of type 1), or −∇^2^ρ_α_ may be positive and −∇^2^ρ_β_ negative with no constraint on their magnitudes (CP maximum of type 2), or it may occur that both −∇^2^ρ_α_ and −∇^2^ρ_β_ are negative, despite −∇^2^s is positive, provided −∇^2^ρ_α_ is smaller in magnitude than −∇^2^ρ_β_ (i.e., | −∇^2^ρ_α_ | < | −∇^2^ρ_β_ |), this latter case being denoted as a CP maximum of type 3. It is worth noting that these different CP maximum types highlight three different physical situations through which a maximum of s can occur. They correspond to three different constraints on the (relative) concentration or dilution of the ρ_α_ and ρ_β_ electron densities. The case of other CP types, saddles and minima, may be similarly analyzed using the information gathered in Table 1.

Regions around CP maxima, where −∇^2^s is positive, behave as sources for the spin density in the remaining parts of the molecule [19,29,30,31] and have the effect of enhancing their α–electron density (α–effect) [19]. Those around CP minima, where −∇^2^s is negative, behave in an opposite way and so they have a β–effect [19]. Regions around CP saddle points may have either an α–effect or a β–effect, depending on whether −∇^2^s is positive or negative [19,29,30,31]. Table 1 shows that the sign of −∇^2^s in a given molecular region, determining whether this region acts as a source or a sink for s, may be realized in 18 different ways corresponding to the 18 CP electron spin density types.

#### 2.1.2. Critical Points of the Electron Spin Density and the Poincaré–Hopf Relationship [11,14,15,16]

The electron spin density, differently from the electron density and likewise the electrostatic potential or the electron density Laplacian, may exhibit either positive (s_+_) or negative (s_−_) local values. The only obvious constraint on s is that its integral over *R*^3^ equals the number of the excess α–electrons. The possibility of negative values for a scalar has an impact on the total number and kind of CPs that may be found for it [35,36,37]. Indeed, negative minima of the scalar may be seen as maxima of the negative of such scalar and so one may topologically study the scalar by separating the regions where it is positive from those where it is negative and treat each of these regions as a distinct system. But special considerations need to be introduced to check whether the number and kinds of CPs fulfil the constraints dictated by topology. In each separate region, regardless the spin density exhibits only positive or only negative values, the number of non-degenerate CPs satisfies the Poincaré-Hopf theorem [11,14,15,16]:Σ_p_(−1)*^I^*^(***X***;*p*)^ = *χ*(*M*)(3)
where the sum runs over all the *p* CPs of the vector field ***X*** bound on the manifold *M* and where *χ*(*M*) is an integer number and a topological invariant, named as the Euler characteristic of the manifold. According to Equation (3),
n_−3_ − n_−1_ + n_+1_ − n_+3_ = *χ*(*M*)(4)
where n_−3_, n_−1_, n_+1_, n_+3_ denote the number of maxima (3, −3), saddle points (3, −1) and (3, +1), and minima (3, +3), respectively. 

By replacing s by –s, the total number of CPs does not change but all signs of CP indexes *I* are reversed. Equation (4) will so read as Equation (5): −n_−3_ + n_−1_ − n_+1_ + n_+3_ = *χ*(*M*)(5)
implying *χ*(*M*) = −*χ*(*M*) = 0 as well as the fulfilment of the relationship given by Equation (6):n_−3_ − n_−1_ + n_+1_ − n_+3_ = 0(6)

Equation (6) holds for all scalar functions on R^3^, including the spin density and the more customarily studied electron density. So, in what way will the spin density differ from the electron density? As shown by Leboeuf et al. [35] and by Gadre et al. [36] for the case of the electrostatic potential, one has to consider the asymptotic behaviour of the investigated scalar function and look at its asymptotic critical points, which are also included in the Poincaré–Hopf theorem relationships (Equations (3)–(6)). Since both s and **∇**s go to zero at infinity, asymptotic CPs do exist, like for the electron density, and enter (Equation (6)) with a negative sign if minima and a positive sign if maxima, Equation (7),
(n_−3_ + n_−_) − n_−1_ + n_+1_ – (n_+3_ + n_+_) = 0(7)
where n_−_ and n_+_ denote the number of asymptotic maxima and minima, respectively. Equation (7) may then be written as Equation (8) to separate the asymptotic CPs:n_−3_ − n_−1_ + n_+1_ − n_+3_ = n_+_ − n_−_(8)

Each positive region extending to infinity has one asymptotic minimum, while each negative region extending to infinity has one asymptotic maximum. For the simpler case of the electron density, Equation (8) will then read as Equation (9) [13]:n_−3_ − n_−1_ + n_+1_ − n_+3_ = nuclei – bond – ring – cages = 1(9)
while, for a spin density having both s_+_ and s_−_ regions extending to infinity, one has to use Equation (8) and count their n_+_ and n_–_ asymptotes, respectively. If the s_−_ regions are fully embedded in s_+_ regions, n_−_ will be equal to zero and the CPs of the whole spin density will fulfil Equation (9), as those of the corresponding electron density (see application to the water molecule in the ^3^B_1_ state in Section 3.1.2).

#### 2.1.3. Spin Maxima and Spin Minima Joining Paths

In the s_+_ regions, each (3, −1) CP links two neighbouring (3, −3) maxima, analogously to the conventional (3, −1) bond critical point of ρ. However, despite the similar function, these (3, −1) CPs bear a different physical meaning. The two unique spin density gradient paths, originating at a (3, −1) CP and ending up at their own (3, −3) CP attractor, are lines of steepest spin density increase and thus paths of maximum spin density relative to any possible lateral displacement. We call their union a *spin maxima joining path*, in formal analogy with the QTAIM bond path of the electron density topology [13]. Likewise, we term as *molecular spin graph* of a molecular system the collection of all its spin maxima joining paths, in close correspondence to the system’s QTAIM molecular graph, which is formed by collecting all the bond paths [13]. What has been said thus far for the s_+_ regions holds true also for the s_−_ regions, where the (3, +3) s minima represent (3, −3) maxima of –s, the negative spin density, and the (3, +1) second–order s saddle points play the role of the (3, −1) spin density CPs for –s. 

So, the union of the two unique spin density gradient paths that have as a terminus the (3, +1) s(**r**) CP and as origin two (3, +3) s(**r**) 3D–repellers may be envisaged as a spin maxima joining path for the –s scalar. We will then have in general spin maxima joining paths in both the s_+_ and the s_−_ regions of the system, associated with the +s and the –s function, respectively. The first ones denote a link between maximally α–polarized centers, while the second ones a link between maximally β–polarized centers. The latter occur in a molecular region where the spin density oppositely polarizes relative to the dominant spin polarization of the system. For the sake of conciseness, we will call these two kinds of lines as α–α and β–β spin maxima joining paths, respectively. Yet, these lines connect 3D extrema, where the β– and the α–component of ρ, respectively, are also, in general, different from zero. Like the collection of the α–α spin maxima joining paths, the β–β spin maxima joining paths also form, collectively, a spin graph. A spin–polarised system may thus be characterized by as many α–α and β–β *molecular spin graphs* as there are numbers of s_+_ and s_−_ separate regions. A formally similar situation occurs for the positive and negative regions of the molecular electrostatic potential [37]. 

Each molecular α–α (or β–β) spin graph will be formed by *m* spin extrema (maxima or minima) joining paths, where *m* is given by half the sum of the maxima (or minima) own *valences v* over the *n* s_+_ and s_−_ separate regions, Equation (10)
(10)$$m=\frac{1}{2}\sum_{i=1}^{n}v_{i}$$

The valence of a spin density maximum (or minimum) is the number of other maxima (minima) it is connected to and it is formally equivalent to the atomic coordination of a QTAIM atom.

It is worth noting that the termini of an α–α molecular spin graph are not in general the nuclei of the system and may be even far off from them. Typically, the termini are in those regions where the singly occupied molecular/natural orbitals are maximally concentrated. Therefore, they normally lie in the QTAIM basin(s) of the paramagnetic atom(s), but not necessarily so. In fact, electron spin is often delocalized on the other atoms of the system, causing other, generally lower in value, spin density maxima to be formed. *A fortiori*, the termini of the β–β molecular spin graphs do not in general correspond to any nucleus of the system. 

#### 2.1.4. Spin Density Basins

Being ω–limits and α–limits of infinite disjoint sets of ∇s trajectories in all directions, maxima and minima of s(**r**) act as 3D–attractors and 3D–repellers, respectively. The space traversed by these trajectories determines the associated basins, which are bounded by local **∇**s zero–flux surfaces (ZFS) S, characterized by **∇**s(**r_s_**)⋅**n**(**r_s_**) **=** 0, **∀**
**r_s_** ∈ S and where **n**(**r_s_**) is the surface outward normal vector at **r_s_**. For an α–spin–polarized system, s maxima and s minima may occur in both the s_+_ and s_−_ regions, though the maxima are preferentially found in the s_+_ and the minima in the s_−_ regions. The basins of the s maxima provide an exhaustive and disjoint partitioning of *R*^3^, which highlights the shape of those regions having, in their interior, a single (3, −3) spin density maximum. The basins of the 3D–repellers yield a different, yet also exhausting and disjoint division of the space. Here, spin density maxima, first–order and second–order saddles lie on the surface borders. Edges and vertices of such borders are α–α spin maxima joining paths and (3, −3) spin density maxima, respectively. This partition thus highlights the molecular α–α spin graph(s) (see Section 2.1.3). There is then a dual description in terms of –s, with the 3D–attractors and the 3D–repellers exchanging their roles. 

All **∇**s zero–flux basins of the spin density, differently from those of the QTAIM theory, may contain points with either positive or negative function values. Furthermore, while the local zero–flux condition of QTAIM basins enable them to have a well–defined kinetic and total energy and thus makes them proper quantum entities [13], no such a fundamental property holds true for the spin density basins. It is therefore of interest to also consider a different kind of spin density basin, namely those enclosing only positive or only negative s values and bounded by s = 0 isosurfaces. We call them s_+_ and s_−_ basins to distinguish them from the conventional zero–flux basins. This alternative, disjoint and exhaustive R^3^ space partition enables us to separate molecular regions in terms of their α– or β–density dominance and, once known the sum of population of s_−_ basins _,_ to have a quantitative measure of the spin counter–polarization effect. By definition, if α–spin is the majority spin, a Restricted Open Hartree Fock (ROHF) wavefunction has s > 0 everywhere and only one s_+_ region. Hence, the location, size and population of the s_−_ basins obtained from less crude approaches give all together a faithful mapping and quantitative evaluation of the spin counter–polarization mechanisms. This is made possible by releasing the α and β–spin orbital equivalence constraint for doubly occupied MOs inherent to the ROHF wavefunction.

## 3. Results and Discussion 

### 3.1. Spin Density Topology of the Water Molecule in the ^3^B_1_ State 

As an illustration of the general aspects of the spin density topology discussed in Section 2, we scrutinize here a simple, yet exemplary molecular case, namely H_2_O in the ^3^B_1_ state. Nuclei are in the (y, z) plane and two unpaired electrons are assigned to two singly occupied (molecular or natural) orbitals (SOMOs or SONOs, hereinafter called SOMOs for the sake of simplicity) dominated by O p_x_–type and by O s, p_z_–type functions, respectively (Appendix A). Details on the applied levels of theory (CASSCF(8,8), hereinafter abbreviated to CAS, UHF, ROHF, all with a 6–311++G(2d,2p) basis set) and on the exploited computational codes are reported in Section 5 and in Appendix A. This global level of theory has been chosen in analogy with that adopted some time ago for the study of the spin density source function on the same molecular system [19]. A 3D representation of the ^3^B_1_ H_2_O electron density and electron spin density, as well as of the densities associated with the two SOMOs (*A*_1_ and *B*_1_) is shown in Appendix A at the CAS and UHF levels. The SOMOs *A*_1_ and *B*_1_ densities, which are equivalent to their spin densities as SOMOs are occupied only by α–electrons, look pretty much the same at all levels, including the ROHF one (not shown). Apart from a slightly greater extension of the s_−_ region, especially close to the O nucleus and on the opposite side of the H nuclei, UHF and CAS spin densities also look qualitatively the same in the molecular plane. However, inspection of the corresponding topologies reveals noticeable differences.

#### 3.1.1. Critical Points of the ^3^B_1_ Water Spin Density

Table 2 and Table 3 report data for all ^3^B_1_ H_2_O CPs (Table 2, maxima and (3, −1) saddles; Table 3, (3, +1) second–order saddles and minima) and for all investigated methods. An identification number N is assigned to each unique CP. The corresponding multiplicity M and type (according to Table 1 classification) are also reported, along with the values of several properties at the CP location. These include the electron spin density s and its magnetic component s_mag_ (see Introduction, Section 1.2 and Section 5), the electron density ρ, the gradient of the α–electron density ∇ρ_α_ (equal to ∇ρ_β_ at the CP, Equation (2)), the Laplacian of the spin density ∇^2^s, its α and β–components ∇^2^ρ_α_ and ∇^2^ρ_β_, and the local Spin Polarization Index SPI(**r**_c_). We define this index as SPI(**r**) = [ρ_α_(**r**)/ρ_β_(**r**)]/(N_α_/N_β_) = [ρ_α_(**r**)/ρ_β_(**r**)]·(N_β_/N_α_), where N_α_ and N_β_ are the system’s α– and β–electron populations. SPI(**r**_c_) is a useful and physically meaningful quantity as it amounts to 1 if the spin density polarization at the CP, expressed by the ratio of the α– and β–electron density values at **r**_c_, equals the ratio of the corresponding global populations (6/4 = 1.5 for ^3^B_1_ H_2_O), while it turns out to be higher (lower) than 1 if the local ratio exceeds (is below) the global ratio. Note that, for ^3^B_1_ H_2_O, a value of s(**r**_c_) = 0, denoting the absence of any spin polarization at the point, corresponds to SPI = 2/3 = 0.667. For the sake of completeness, eigenvalues μ_i_ (i = 1–3) of the Hessian matrix of ∇^2^s are also listed in the Appendix A. 

Save the ROHF case, all considered methods predict the same number of unique CPs, 3 maxima, 4 (3, −1) saddles, 6 (3, +1) second–order saddles and 2 minima. Taking into account the CP multiplicities (Table 2 and Table 3), there are five maxima, 8 (3, −1) saddles, 8 (3, +1) second–order saddles and four minima for a total of 25 CPs. The ROHF α = β constraint, for the doubly occupied orbitals, implies the absence of s_−_ regions in its spin density distribution and a simpler topology, with 5 maxima and 8 (3, −1) saddles as for the other wavefunction models, but with only four (3, +1) CPs (N = 8 and N = 9, both having M = 2) and no minima. There are so only 17 spin density CPs at the ROHF level, two of which (N = 9) are lacking in the other two models. 

Overall, the spin density topology is richer and less trivial than that of the electron density which describes the ^3^B_1_ H_2_O molecule in terms of just three maxima (almost) at the nuclear positions and two (3, −1) O–H bond critical points. Figure 1a and Figure 2a show CAS electron spin density contour plots in the two σ_v_ planes of water, namely the σ_v_ plane of the three nuclei (y, z), Figure 1a, and the σ_v’_ plane perpendicular to it (x, z), Figure 2a, containing only the O nucleus. All CPs are cumulatively shown on these plots, the unique ones being labelled by their identification number N. The four CPs 6 do not lie on either σ_v_ or σ_v’_ and their locations are all projected on each of these two planes (each projection accounts for two CPs 6). Figure 1b and Figure 2b display the CAS ∇^2^s plots in the same planes as those of Figure 1a and Figure 2a, respectively, and, for the sake of better appreciating data in Table 2 and Table 3, also show spin density CPs locations and labels. 

Similar plots for the other wavefunction models are shown in Appendix A (UHF) and Appendix A (ROHF). In addition, Appendix A display the differences between CAS and UHF spin densities s (Appendix A), or between CAS and UHF ρ_α_ (Appendix A) or ρ_β_ (Appendix A) electron densities.

We start our analysis by considering the (3, −3) maxima and (3, −1) CPs (Table 2), which, as we said, are all topologically equivalent for the three studied wavefunctions. 

• (3, −3) CPs

There are two equivalent s maxima (N = 1) located for all models at 0.19 Å above and below the nuclear plane and practically right above/below the oxygen nucleus. They correspond to spin density maxima associated with the B_1_ SOMO (dominated by O p_x_–type functions and minor contributions from H p_x_–functions). The other two equivalent maxima (N = 2) are located in the molecular plane, along the O–H axes, behind the H nuclei. They are related to the A_1_ SOMO (dominated by the O s–type and p_z_–type functions and with contributions from the H s–type and p_y_–, p_z_–functions). The last maximum (N = 3) lies in the molecular plane, almost coincident with the O nucleus, and it is also associated with this latter SOMO. Its proximity to the O atom is immediately revealed by the large value of ρ at the CP, while the location of CP 2 close to the H atom and far from the region of the paramagnetic O atom, is manifest from an s value (s = 0.012, CAS level) lower by two orders of magnitude relative to those at the other two unique maxima.

The spin density at CP 1–3 is dominated by the s_mag_ component, as all the maxima are associated with SOMO’s. The way such dominance is realized depends on the kind of maximum and of model wavefunction. While in ROHF, by definition s_mag_ = s at all maxima, for the UHF and CAS wavefunctions, s_mag_ occurs to be slightly (CP 1) or more significantly (CP 3) smaller than the value of s at these CPs or even larger (CP 2) than it. Such different behaviour reflects the different ways the relaxation effects play their role in the various molecular regions. For the ROHF model, the CP condition given by Eq. 2 implies ∇ρ_mag_ ≡ ∇ρ_αmag_ = 0, i.e., the spin density topology is only determined by the ρ_mag_ ≡ ρ_αmag_ topology (since each ROHF doubly occupied orbital satisfies Equation (2) everywhere). The relative distance of a given method spin CP from the corresponding ROHF spin CP, $d_{ROHF}^{method}$, provides another measure of the importance of spin relaxation effects. Appendix A lists such $d_{ROHF}^{UHF}$ and $d_{ROHF}^{CAS}$ distances, for all CPs common to the three methods (CPS 1–8, Table 2 and Table 3). It is seen that such distances are, in general, small, below or well below 0.05 Å, but are occasionally far larger. Locations of maxima 1 and 3, those characterized by large s and s_mag_ values, are almost coincident for the three methods, while significant displacements, as large as about 0.20 Å (UHF) and 0.26 Å (CAS) relative to ROHF, are observed for the two maxima, CP 2, along the O–H axes. Large displacements also occur for CPs 6 and 8 ($d_{ROHF}^{UHF}$: 0.13, 0.08 Å; $d_{ROHF}^{CAS}:\;0.18,\;0.13\;Å)$ as these ((3, −1)) saddle and (3, +1) ring CPs are strongly related to the position of maximum 2 (*see infra*). Enabling spin relaxation and the consequent occurrence of a s_−_ region has a decisive impact on the position of the CP 2 maxima, which shift (far) behind the H nuclei, yet still along the O–H axis (CAS, Figure 1a; UHF, Appendix A and ROHF, Appendix A). Besides the already discussed measures ($s_{mag}/s\;\mathrm{and}\;d_{ROHF}^{method}$), spin relaxation effects are also clearly evident from the differences, relative to the ROHF model, in the s, ∇ρ_α_ , ∇^2^s, etc., values. The higher the wavefunction quality, the larger the differences, even though the UHF estimates are in general much closer to CAS than to ROHF results (compare Figure 1 and Figure 2 with Appendix A, UHF, and Appendix A, ROHF). 

The three unique spin density maxima may be distinguished also by their SPI value and type (Table 2). SPI is significantly larger than one at maximum 1, while being lower than one at the other two maxima. In particular, CP 3 has SPI ≈ 2/3, the value denoting the absence of local spin polarization, despite its highest s value. Indeed, CP 3 has the highest CP spin density, but the lowest s(**r**_c_)/ρ(**r**_c_) ratio *ra* (CAS: *ra* = 0.339, 0.167 and 0.002, for CP 1, 2 and 3, respectively), as it lies almost on the O nucleus, in the innermost O core region, which is expected to be hardly spin–polarized. On the other hand, the large SPI deviation from 1 for CP 1 is a clear sign of the dominance of SOMO *B*_1_ at that location, while the SPI value slightly lower than one (UHF and CAS) for CP 2 denotes its being involved in the spin delocalization mechanism from the paramagnetic center to the H atoms and the fact that the greater part of spin polarization remains on the O atom. This latter feature is particularly evident in the ROHF model, having an SPI value of only 0.765 at CP 2, so (much) closer to 2/3 than to one. 

As for spin density maxima types, we observe (Table 2) that the maximum 1 is of type 3 for all computational models, α– and β–densities being both charge depleted at CP. Yet, the β–density is more depleted then the α–one. This results in a spin density concentration at this CP, as required by its nature of maximum. In our previous study [19], we considered the –∇^2^ρ (3,+1) CPs located above and below the O nucleus and lying 0.37 Å away from the nucleus, i.e. twice as far from the O nucleus than CP 1, as representative of the unpaired electron distribution in the O p_x_ orbital. At the −∇^2^ρ (3, +1) CP, the α–density is concentrated, but the β–density is twice as much depleted, a feature denoting a type–2 spin density CP. Despite the CP 1 and the −∇^2^ρ (3, +1) CP are both characterized by a spin density concentration, there is clearly a profound difference in the physical mechanism leading to such concentrations. Moreover, the −∇^2^s magnitude at the former CP is six times larger than at the latter. The other large s spin density maximum 3 is of type 1 for all computational models. At this CP, both α– and β–densities are concentrated, yet the α–density more than the β–density is, yielding, as required by the CP nature, an overall spin density concentration at the CP. Not unexpectedly, the only CP type difference among the considered methods is found for the low s(**r**_c_) spin density maximum (CP 2). Being related to spin delocalization mechanisms, its type is largely affected by the computational method. It is of type 1 at the ROHF level and of type 2 and 3 at UHF and CAS levels, respectively. So, all possible cases are covered by the three adopted methods: α– and β–densities both charge concentrated (with α–density more than β–density) at the ROHF level, α– and β–densities, respectively, concentrated and depleted at the UHF level and α– and β–densities both charge depleted (but with α– less than β–density) at the CAS level. 

• (3, −1) CPs

The four unique (3, −1) saddle points correspond to the minimal Spin density values along the 8 α–α spin maxima gradient paths (see infra, Section 3.1.3 and Figure 3) connecting: (a) the spin density maximum 3 with each spin density maximum 1 (saddle 4, M = 2); (b) the two spin density maxima 1 (saddle 5, M = 1); (c) each spin density maximum 1 with each spin density maximum 2 (saddle 6, M = 4); and (d) the two spin density maxima 2 (saddle 7, M = 1). Spin density values at these (3, −1) CPs denote the nature of the maxima they are linked to. CPs 4 and 5, connected through their associated spin maxima joining paths to maxima both located in the QTAIM basin of the paramagnetic center, have quite large s(**r**_c_) values [> 0.1 e^−^(bohr)^−3^ ]. Those involving either one maximum (CP 6) or two maxima (CP 7) hosted in the basins of the H atoms, where the spin excess is partly delocalized (<30%, ρ ZFS, see Section 3.1.4), have instead a s(**r**_c_) value that is two orders of magnitude lower. All CPs have a s(**r**_c_) value dominated by the magnetic component. However, for CPs 4 and 5, bearing the largest s(**r**_c_), this dominance decreases significantly on passing from ROHF (100% by definition), to UHF (92% and 77%, respectively) and eventually to CAS (88% and 69%). The SPI is far below one for all (3, −1) saddles, except for the saddle between the two maxima hosted in the basins of the H atoms (CP 7). This CP has the highest SPI (around 2.5) and the lowest s(**r**_c_) value among (3, −1) saddles, in accordance with both the spin density delocalization from the paramagnetic atom to that region and the somewhat limited relevance of how this mechanism takes place. Interestingly, the various (3, −1) CPs all differ in type, save CPs 5 and 6 that are both of type 6, while all considered methods agree in predicting the same CP type for a given (3, −1) CP. Differently from the s maxima, the (3, −1) CPs may have either positive or negative −∇^2^s(**r**_c_) value. At CPs 5 and 6 the spin density is depleted, being both composing densities depleted, yet the α–density more than the β–density. These CPs are thus both of type 6. At the other two CPs, the spin density is concentrated, but the CPs belong to two different types. The CP 4 is of type 1, while CP 7 linking the spin maxima located just behind the H nuclei is of type 3. At CP 4, both the α– and β–density are concentrated, the α–density more so than the β–density, while, at CP 7, the situation is quite different, as both composing densities are diluted but the α–density is less diluted than the β–density. 

In the case of the electron density topology, the sign of the electron density Laplacian at the (3, −1) bond critical point (bcp) has been associated with the nature of the bonding interaction [13], this latter being shared (or covalent) for a negative Laplacian and, conversely, not shared (or closed–shell like) for a positive Laplacian. This classification holds true for bonds where both the involved atoms exhibit a valence shell charge concentration when in isolation [11]. One is therefore wondering whether a spin concentration or dilution at a spin density CP also retains any specific chemical meaning. CPs 5 and 6, where the spin density is diluted, have a much larger magnitude for the curvature μ_3_ along their associated spin maxima joining path than for any of the curvatures μ_1_ and μ_2_ associated with their interbasin surface (see Appendix A). This denotes a marked tendency of the unpaired electrons associated with the spin density maxima that these CPs are linked to to remain isolated and peaked. Not surprisingly, this situation occurs only when the maxima 1 associated with the unpaired electrons in the B_1_ SOMO are involved. To be correct, CP 4 is also linked to maximum 1 (and to maximum 3). However, being located very close to maximum 3, which has a clear spherical distribution (see μ values in Appendix A), its μ_3_ curvature has a magnitude that is slightly smaller or only marginally larger than the perpendicular curvatures μ_1_ and μ_2_ which so prevail in determining a negative Laplacian value at the CP. The more or less “shared” nature of spin maxima interactions may be also judged in terms of the ratio r = s(3, −1)/s(3, −3)_av_ where s(3, −3)_av_ is the average of the Spin density values at the two linked maxima, though the length of the spin maxima joining path also matters. In general, the value of *r* will be large for the shared interactions (−∇^2^s (**r**_c_) > 0) and definitely smaller for the not shared ones (−∇^2^s(**r**_c_) < 0). Its value amounts to about 0.57–0.50 (ROHF and UHF, 0.57; CAS 0.50) for spin maxima joining paths associated with CP 4 and to 0.17–0.25 (ROHF, 0.17; UHF, 0.20 and CAS, 0.25) for that associated with CP 7, while it significantly lowers to 0.02–0.01 (ROHF and UHF, 0.02; CAS, 0.01) for those related to CP 6 and to 0.16–0.24 (ROHF, 0.16; UHF, 0.21 and CAS, 0. 24) for that related to CP 5. Despite being a “shared” interaction, the *r* value for the spin maxima joining path associated with CP 7 is small due to its exceptional length of 3.11 Å (CAS), to be compared with the values of 0.19, 1.88, 0.70 Å (CAS) for the spin maxima joining paths involving CPs 4, 6 and 5, respectively (see Appendix A for the complete set of spin maxima joining paths lengths data and Figure 3 and Figure 4 in Section 3.1.3 for a pictorial representation of these paths at the CAS and UHF levels, respectively).

Another interesting chemical feature is revealed by the location of the (3, −1) CP along the spin maxima joining path. With respect to the mid–point of such a path, the (3, −1) CP displaces towards the maximum having the lower spin population (Section 3.1.4, Table 4) between the two linked maxima. Such a displacement is conveniently quantified by the percentage shift Δ% defined as Δ% = [Δ_(3, −1)_ × 100]/(0.5 *l*), where Δ_(3, −1)_ is the (3, −1) CP shift from the line midpoint and *l* is the line length [38]. Apart from the trivial case of the lines linking homo–maxima, where symmetry dictates Δ% to be equal to zero, the CPs 4 are significantly shifted (Δ% = 29.5, CAS) towards the maximum at the O nucleus and the CPs 6 are less markedly (Δ% = 14.7, CAS) shifted towards maxima 2, located close to and behind the H nuclei. The spin populations of the small basin centered on the O nucleus and of the basin associated with the H nucleus are (CAS) 0.005 and 0.394 electrons to be compared with the (much) larger spin population value of 0.604 (CAS) for the basins of the O atom unpaired electrons (Section 3.1.4, Table 4). The larger the spin population of a basin, the bigger the basin volume is and the farther the (3, −1) CP(s) on the basin bounding surface are displaced away from the basin (3, −3) CP attractor. This is reminiscent of the heterocovalent bonding interaction, where the electron density (3, −1) CP is displaced towards the less electronegative atom.

•  (3, +1) and (3, +3) CPs

We now discuss the (3, +1) and (3, +3) CPs patterns (Table 3 and Figure 1 and Figure 2 and Appendix A). As mentioned earlier, the ROHF model topology greatly differs from the UHF and CAS models for these specific CPs kinds, while the UHF and CAS topologies, though different, are more similar to each other and retain the same numbers of (3, +1) and (3, +3) CPs. Common to all wavefunction models are the CPs 8 (M = 2) (Table 3 and Figure 2), associated with the spin density minimum in the rings (Figure 3) formed by a maximum 1 and two maxima 2. All models describe them as of type 6 (3, +1) CPs, with positive ∇^2^s value at the CPs and the α–density more diluted than the β–density. Although it is not a necessary consequence of the lack of s_−_ regions, the ROHF model does not predict any minimum (Table 3 and Appendix A). The only two additional ROHF (3, +1) CPs, labelled as 9 (Appendix A), are associated with the rings formed by joining maxima 1–3–2–1–3 (Figure 3 and Appendix A). All other (3, +1) and (3, +3) CPs pertain only to the UHF and CAS models and the differences between the two models essentially arise from the noticeable difference of their s_−_ regions. 

Comparison of Figure 1a, Figure 2a (CAS) and Appendix A, Appendix A (UHF) highlights the spin counter–polarization effects; hence, the size of s_−_ regions are both largely overestimated by the UHF model (assuming that the CAS estimate is the more reliable). In particular, the s_−_ region, instead of being essentially confined between the O and the H nuclei and between the H nuclei, significantly extends itself far behind the O nucleus in the UHF model (Appendix A). A close look to the (CAS–UHF) s, ρ_α_ and ρ_β_ density differences (Appendix A) reveals that the CAS higher spin density around the oxygen nucleus and the CAS lower spin density around the H nuclei in the molecular plane result from a complex and opposite ρ_α_ and ρ_β_ electron density rearrangement. In the CAS model, both α and β electron densities are deprived around the O nucleus, relative to UHF, but the β density more than the α one, thus yielding, there, an overall spin density increase. On the other hand, relative to UHF, both α and β electron densities are increased in the CAS model around the H nuclei, yet the latter more than the former, resulting, there, in an overall spin density decrease. Both mechanisms concur in reducing the spin counter–polarization effects at the CAS level, relative to UHF.

The (3, +1) CPs 10 and 11, both located in the s_+_ region, are common to CAS and UHF models. They are related to the 4MR (four–membered ring) ring joining 1–2–1–2–1 maxima and the 3MR (three membered ring) joining 3–1–1–3 maxima, respectively (Figure 3 and Figure 1 and Figure 2 (CAS) and Appendix A (UHF)). They both have M = 1, as they need to be located on the C_2v_ axis by symmetry and are thus visible in both σ_v_ and σ_v’_ planes. The CP 11 has the highest s value among all (3, +1) CPs being related to a 3MR formed by the highest spin density maxima (CP 1 and CP 3). Such maxima are quite peaked since the s value at the ring CP, despite being far larger than at other (3, +1) CPs, is still two orders of magnitude lower than at the ring vertices. For the UHF, model both the remaining (3, +1) CPs 12–14 and the (3, +3) minima 15–16 all fall in the s_−_ region. For the sake of simplicity, they may be seen as (3, −1) saddles and (3, −3) maxima of the negative spin density, −s. These points lie in the σ_v_ plane and are thus all visible in Appendix A. They form a 4MR with 16–16–15–15–16 as –s peaks and 13, 14 and 12 as –s (3, −1) saddles, linking the 16–16, 16–15, 15–15 –s peaks, respectively (see Appendix A and the spin extrema joining paths for the UHF negative spin density shown in the right panel of Figure 4 in Section 3.1.3). The spin density magnitude at –s maxima is quite small (0.002–0.007 e^−^(bohr)^−3^) and that at the –s (3, −1) CPs is necessarily smaller. Despite these –s maxima are linked to form a 4MR, there is not a corresponding –s (3, +1) ring. Indeed, the s_−_ region is fully embedded in a s_+_ region, hence its boundaries are the s = 0 envelopes, whose points collectively act as a –s function minimum. The CAS model predicts just one unique s minimum (CP 15, M = 2) and two s (3, +1) saddles, 12 and 13, all having negative s values (note that the s value at CP 13 is vanishingly small, being −1 × 10^−5^ e^−^(bohr)^−3^). Again, using for the sake of simplicity the –s scalar, a ring chain is formed where the two –s maxima 15 are linked through two different spin extrema joining paths having CPs 12 and 13 as –s (3, −1) saddles (see Figure 1 and the spin extrema joining paths for the CAS negative spin density shown in the right panel of Figure 3 in Section 3.1.3). Analogously to the UHF model and for the same reason discussed for that model, also at the CAS level there is not an associated –s ring point. Finally, there are (Figure 1) two (3, +3) s minima (CPs 16) and two (3, +1) s CPs (CPs 14), all characterized by relatively high positive Spin density values (0.020 and 0.023 e^–^(bohr)^−3^, respectively). They are associated, respectively, to two cage CPs inside two quite small volumes bounded by the central 3MR 3–1–1–3 and to two external 3MRs each enclosing one cage CP 16. They are constrained by symmetry to lie in the σ_v_ plane (Figure 1). As reported earlier, these CPs 16 and 14 do play a different role in the UHF model as they belong both to the s_–_ region. 

#### 3.1.2. Critical Points of the ^3^B_1_ Water Spin Density and Poincaré–Hopf Relationships

Since the s_−_ region of the ^3^B_1_ H_2_O molecule is fully embedded in the s_+_ region extending to infinity, the whole spin density of this molecule should fulfil Equation (9) (Section 2.1.2). On the other hand, the CPs of the s_+_ and s_−_ regions, taken separately, should fulfil Equation (9) (one asymptotic minimum) and Equation (7) (with both n_+_ and n_–_ equal to zero), respectively. All such information may be retrieved from the data in Table 2 and Table 3, discussed in the previous Section 3.1.1. Both the total UHF and CAS spin densities recover n_–3_ = 5, n_–1_ = 8, n_+1_ = 8, n_+3_ = 4; hence, 5 –8 +8 –4 = 1, Equation (9), and the Poincaré–Hopf theorem turns out to be satisfied. Analogously, for the ROHF model, having n_–3_ = 5, n_–1_ = 8, n_+1_ = 4, n_+3_ = 0, Equation (9), i.e., 5 – 8 + 4 – 0 = 1, is fulfilled. As discussed above, the CPs of the s_+_ regions also fulfil Equation (9) for all models (UHF: n_–3_ = 5, n_–1_ = 8, n_+1_ = 4, n_+3_ = 0, 5 – 8 + 4 – 0 = 1; CAS: n_–3_ = 5, n_–1_ = 8, n_+1_ = 6, n_+3_ = 2, 5 – 8 + 6 – 2 = 1; ROHF: see above), while those of the s_−_ regions (only UHF and CAS) fulfil Eq. 7 (UHF: n_–3_ = 0, n_–1_ = 0, n_+1_ = 4, n_+3_ = 4, 0 –0 + 4 –4 = 0; CAS: n_–3_ = 0, n_–1_ = 0, n_+1_ = 2, n_+3_ = 2, 2 – 2 = 0). Though the satisfaction of the Poincaré–Hopf theorem only provides a necessary condition [11], we are quite confident to have retrieved all the existing CPs for all investigated wavefunctions, due to the very careful CP searches we performed.

#### 3.1.3. Spin Maxima and Spin Minima Joining Paths and Molecular Spin Graphs of the ^3^B_1_ Water Molecule

The ^3^B_1_ H_2_O molecule, having both s_+_ and s_–_ regions at the UHF and CAS levels, is described by two molecular spin graphs, one α–α and one β–β (Figure 3, CAS and Figure 4, UHF). While α–α spin graphs are topologically equivalent and very much alike in shape at the CAS and UHF levels, their β–β graphs are topologically different, as dictated by the different topology of the two models in the s_–_ region (see Section 3.1.1). Their comparison provides a pictorial representation of the different description of the spin polarization mechanisms by the two wavefunction models. Due to the α ≡ β constraint on doubly occupied orbitals, the ROHF model describes the ^3^B_1_ H_2_O molecule with *only* the α–α molecular spin graph, closely resembling the α–α graphs of the CAS and UHF models. The lack of flexibility of the ROHF model has thus an immediately visible impact on the overall spin structure of the molecule. Interestingly, the α–α graphs are not only alike for all the three models, but also similarly dominated by s_mag_. As an example, Appendix A clearly shows that the CAS s_mag_ α–α graph is topologically equivalent to the total spin density α–α graph (Figure 3) and may be hardly distinguished from the latter except for minor details around maxima CP 2.

The α–α molecular spin graph neatly differs from the (electron density) molecular graph (Figure 3 and Figure 4). The spin structure of the ^3^B_1_ H_2_O molecule is more complex than its total electron structure, *but both electron and spin structures concur to a complete description of the molecule.* It is like observing the same molecule by wearing two different pairs of lenses. One of these pairs is working in summation of the α– and β–densities, while the other in subtraction of the same densities. The spin density structure does not have a direct connection, namely a spin maxima joining path, between the O nucleus and the H nucleus as in the molecular graph. Yet, it exhibits a direct link between the maxima located nearby the H nuclei, which is totally absent in the molecular graph. The two O–H linkages of the conventional structure of water are replaced in the spin graph by four much longer linkages between maxima associated with the unpaired electrons of the O atom, lying quite far apart from the nucleus (see Section 3.1.1), and the maxima close to the H nuclei. In addition, there are three more gradient paths, all contained within the O atomic basin, which are totally absent in the ρ(**r**)–based structure of the ^3^B_1_ H_2_O molecule. One of these paths connects the two unpaired electron maxima, while two more lines connect each unpaired electron maximum to the maximum located on the O nucleus. While the conventional (chemical and topological) valencies of the O and H (3, –3) attractors are 2 and 1, the maxima of the spin density turn out to have higher valencies (Equation (10)). The α–α graph has five vertices, eight spin maxima joining paths; the valency of the maximum located on the oxygen nucleus (CP3) amounts to 2, those of the maxima associated with the O unpaired electrons (CP 1 ) are as big as 4 and those of the maxima close to the H nuclei (CP 2) amount to 3. The lengths and Δ% values for all unique α–α spin maxima joining paths are listed in Appendix A for all investigated models. We have already discussed part of these data in Section 3.1.1, in relation to the nature of the interacting maxima. Here, we supplement that discussion by noting that UHF predict lengths and Δ% values, which are intermediate between those predicted by the ROHF and CAS models though, in general, much closer to the reference values provided by CAS.

#### 3.1.4. Spin Density Basins of the Water Molecule in the ^3^B_1_ State

Several basin properties for various basin definitions and wavefunction models are reported in Table 4 for the the ^3^B_1_ H_2_O molecule. The kinds of considered basins are: (a) those defined through the catchment regions of the ρ and s scalar fields (3, −3) attractors and through their associated local zero–flux boundaries (∇ρ and ∇s ZFSs, respectively) and (b) also those bounded by s = 0 isovalue surfaces and including in their interior either only s_+_ regions and one (3, −3) s CP or only s_−_ regions and one (3, +3) s CP (see Section 2.1.4). The basins defined through the spin density are labelled in the Table 4 as Ω_x_ where x denotes the numeric label of their (3, −3) s maximum (Table 2) or, for s_−_ regions, (3, +3) s minimum (Table 3). All basins in Table 4 are characterized by their multiplicity M, by their electron and (excess) spin populations, N_Ω_ and SP_Ω_, and by the magnetic component of this latter population, SP_mag,Ω_. The basin size is quantified through the basin volumes V1_Ω_ and V2_Ω_. For all basin types, V1 and V2 are defined in terms of electron density values ρ and namely as the volumes of those basin regions where ρ is equal to or exceeds 0.001 or 0.002 e^−^(bohr)^−3^, respectively [13].

Analogously to the local spin polarization index (see Section 3.1.1), we define a basin spin polarization index, SPI_Ω_, as SPI_Ω_ = (N_α,Ω_/ N_β,Ω_) × (N_β_/N_α_) where N_α,Ω_ and N_β,Ω_ are the α– and β–electron populations of basin Ω and N_α_ and N_β_ the corresponding quantities for the total system. The integral spin polarization index clearly retains the same interpretation as its local version (see Section 3.1.1), the only difference being that it refers to a spin polarization feature of a molecular region, rather than to this same feature at a specific point of the molecule. Besides the quantities SP_Ω_, SP_mag,Ω_ and SPI_Ω_, the basin Ω average spin density ${\bar{s}}_{\Omega }$, ${\bar{s}}_{\Omega }=SP_{\Omega }/V1_{\Omega }$, provides another important index for a proper characterization of the spin polarization nature of a basin. While SPI_Ω_ yields a measure of the average α–electron excess in the basin relative to this same excess in the whole system, ${\bar{s}}_{\Omega }$ is an absolute measure of the basin average α–electron excess. Thus, differently from SP_Ω_ and SP_mag,Ω_, both SPI_Ω_ and ${\bar{s}}_{\Omega }$ are intensive properties. Yet, while SPI_Ω_ may be lower or greater than 1 regardless of the basin size, ${\bar{s}}_{\Omega }$, for a given spin population SP_Ω_, will be small or large depending on this size.

With regards to the conventional QTAIM atomic partitioning (**∇**ρ ZFS) [13], the O atom has at the CAS level a net negative charge of 0.74 e^−^ and an excess spin population of 1.41 e^−^. About 0.59 e^−^ of the α–spin electron excess of two is thus transferred from the paramagnetic center to the H atoms. The spin populations are dominated by their magnetic component SP_mag,Ω_, which in the case of the H atom is even surpassing by 0.01 e^−^ (0.306 e^−^) its total spin population value of 0.296 e^−^. The basin Ω average spin density, ${\bar{s}}_{\Omega }$, is twice as large for the O atom (0.008 e^−^) as for the H atom at all wavefunction levels, but its spin polarization index SPI_O_, slightly lower than one, is less than half the SPI_H_ value, which is quite high and close to 2. Relative to CAS, such a difference is magnified at the UHF and even more so at the ROHF level. From these numbers, we learn that the spin density is twice as concentrated in the basin of the paramagnetic atom, but also that the dominant α–spin density is prevailing over the minority β–spin density more in the H basins than in the O basin. Furthermore, the enhanced prevalence of α–spin density in the H basin is overestimated by the UHF and, particularly, by the ROHF model, relative to the reference CAS model.

Partitioning the R^3^ space through the **∇**s ZFS boundaries, rather than through the **∇ρ** ZFS ones, increases the basin number from 3 (2 unique) to 5 (3 unique) (Table 4). A pictorial representation of these five basins at the CAS level is shown in the top panels of Figure 5. The Ω_3_ basin surrounding the O nucleus, despite being very small (V1 = 0.02 (bohr)^3^, CAS), has a significant electron population (0.935 e^−^, CAS), being in the O core region. For this same reason, its spin population is negligible (0.005 e^−^, CAS) and its SPI_Ω_ value (0.674, CAS) is very low, but its ${\bar{s}}_{\Omega }$ value is by far the highest one. It is about 20 and 60 times larger than the values in the Ω_1_ and Ω_2_ basins, respectively. All these values for the Ω_3_ basin refer to the CAS level, yet the picture they provide is qualitatively similar to that predicted by the less accurate UHF and ROHF models. The two Ω_2_ basins are each about 1.5 times as big as the H basin (**∇**ρ ZFS) and are the **∇**s ZFS boundary basins with the largest size, being (Table 4 and Figure 5) more than twice as big as Ω_1_, in terms of V1 values. Indeed, Ω_2_ basins enclose the whole molecule and also recover a large region of the **∇**ρ ZFS oxygen basin. As a consequence, the electron and spin populations of the two Ω_2_ basins sum up to 1.826 and 0.788 e^−^ (CAS values). Both values are about one and a half times the size of the corresponding estimates for the two **∇**ρ ZFS H basins (1.256 and 0.592 e^−^). On the contrary, summing up the electron and spin populations of the two Ω_1_ basins and of their enclosed Ω_3_ basin yield 8.175 and 1.213 e^−^ (CAS values), to be compared with 8.743 and 1.406 e^−^ for the **∇**ρ ZFS O basin. Apart from such differences, the Ω_1_ and Ω_2_ basins resemble the O and H basins, respectively, in terms of their intrinsic nature. This is testified by correspondingly similar intensive properties ${\bar{s}}_{\Omega }$ and SPI_Ω_. 

Finally, partitioning the R^3^ space in terms of s_+_ and s_−_ regions each including one (3, −3) and one (3, +3) CP, respectively, increases the number of basins from 5 (3 unique) to 7 (4 unique) relative to the **∇**s ZFS partitioning (Table 4, CAS level). A pictorial representation of these 7 basins is shown in the bottom panels of Figure 5. At the ROHF level, the **∇**s ZFS and s = 0 isovalue surface boundaries partitioning obviously coincide, while at the UHF level there are two more s_−_ regions (Ω_16_) with respect to CAS, which implies a total of nine basins (five unique), four of them being associated with s_−_ regions. The electron and (α–β) spin population of all s_−_ basins sum up to 0.424 and −0.006 e^−^ at the CAS level and to 0.540 and −0.004 e^−^ at the UHF level. The spin populations of these basins are dominated by the relaxation component (see Section 1.2), SP_mag,Ω_ being, by nature, positive. The V1 volume associated with all s_−_ basins is small (2.9 and 3.4 (bohr)^3^, at the CAS and the UHF level) as is their average spin density ${\bar{s}}_{\Omega }$ (−0.002 (bohr)^−3^). The spin polarization index SPI_Ω_ needs to be smaller than 2/3, but it is just below this value (about 0.65); hence, it is marginally different from the ρ_α_ = ρ_β_ reference value (2/3). Electron and electron spin populations, sizes and other properties of the s_−_ basins collectively highlight the relevance of going beyond the ROHF approximation and restriction. As reported above, about 6% of the valence electrons in the molecule are included in these basins. All other basins retain, at a qualitative level, the properties of the corresponding basins of the **∇**s ZFS boundary basins.

#### 3.1.5. Source Function Reconstructions

A full source function (SF) reconstruction at all spin density CPs and using as sources those calculated in the basins bounded by the **∇**s and **∇**ρ ZFSs was performed for all the three considered models. For the sake of conciseness, only data at the CAS level and for the (3, −3) and (3, −1) CPs are listed in Table 5. Indeed, UHF results resemble qualitatively the CAS ones, while those at the ROHF level differ, particularly in the case of CP 6, due to its displaced location and to the different portrait of the ROHF spin density Laplacian in the regions around the O atom and CP 6, relative to CAS and UHF models. 

The source function percentage contribution [41] from each basin Ω is listed in the corresponding row and labelled as SF%_CPX_, where x denotes the specific (3, −3) or (3, −1) spin density CP it refers to (Table 2). When a CP is differently distant from the attractor of a unique basin Ω_y_ having a multiplicity M > 1 (Table 4), the farther basin in Table 5 is the primed one (Ω_y_’). Considering first the **∇**s ZFS bounded basins, we notice that the spin density at the (3, −3) attractors is dominated by the SF contribution of their own catchment basin. This contribution exceeds 94% for basins Ω_1_ and Ω_2_, while it is somewhat smaller (83.5%) for the basin Ω_3_ centered on the O nucleus where the two nearby, embedding and much larger basins Ω_1_ contribute for the remaining 16.5%. SF contributions at the (3, −1) spin density CPs are clearly (far) more delocalized. The spin density at the saddle between the two Ω_1_ attractors (CP 5) is fully explained by the SF% contributions of the two joined basins (100.80%), while the spin density at the saddle (CP 4) between Ω_1_ and Ω_3_ attractors is dominated by the SF% contribution of the larger and more spin–populated basin Ω_1_ (61.8%). The other linked basin, Ω_3_, contributes with 27.4% and the very close Ω_1_’ basin the remaining 10.7%. In summary, SF contributions to saddles 4–5 are very local and related to the sources from the basins one may associate to the O atom. Similarly, the spin density at the saddle (CP 7) between the Ω_2_ attractors is dominated (94.4%) by the sources of their two associated basins. The four saddles 6, linking Ω_1_ and Ω_2_ attractors, are different. The spin density Laplacian at CP 6 is positive, and this saddle is largely displaced from the spin maxima joining path midpoint towards the Ω_2_ attractor. It denotes an asymmetric interaction and, as discussed in Section 3.1.1, also a “not–shared” one. Contributions from the two linked attractors are therefore quite different, Ω_1_ even causing a negligible removal of spin density at the saddle (SF% = −0.1), Ω_2_ providing the dominant contribution (51.6%) and the farther Ω_1_’ contributing positively rather than negatively as Ω_1_ and noticeably (36.9%). Moreover, the source from the very far Ω_2_’ basin, though by far lower than that of the linked basin Ω_2_, is not negligible (7.8%). SF analysis confirms the peculiar nature of the Ω_1_–Ω_2_ interaction and describes it as fairly delocalized. Spin density reconstructions at CPs 1–7 may also be appreciated in terms of sources from the conventional O and H QTAIM basins (Table 5). CPs 1 and 3–5 are located within the O basin with this latter contributing with over 99% of the density. CP 2 is even overdetermined (121.2%) by the source of the H basin where it is located, H’ also giving a positive contribution (12.8%) largely compensated for by the negative O basin source (−34.0%). The delocalized, asymmetric, and not shared nature of CP 6 is confirmed by the overdominating H contribution (123.1%), the opposing source contribution from the O (−62.5%) and the significant concurring contribution from the farther H’ basin (39.5%). Finally, saddle 7 is largely overdetermined by the sources of the two H atoms (175.2%), which oppose the large negative O atom source (−75.3%).

## 4. Materials and Methods

As in our previous study of the ^3^B_1_ H_2_O molecule [19], quantum mechanical calculations were performed in vacuo by means of the Gaussian09 program package [42], using the CASSCF (8, 8), UHF, ROHF levels of theory and a 6–311++G(2d,2p) basis set. Geometries were optimized and taken at the ROHF and UHF spin–contamination annihilated levels, for the ROHF and UHF wavefunctions, respectively, (O–H bond length and HOH angle negligibly differ by only 0.001 bohr and 0.3 degrees between the two computational levels). CASSCF computations were performed at the UHF optimized nuclear structure and using as starting guess the UHF spin contamination annihilated natural orbitals. Natural orbitals were obtained through the pop = no option and the two magnetic orbitals were recognized, based on their occupation numbers. These latter are by definition equal to 1 for the ROHF model and resulted either equal to 1 or marginally different from this value (highest deviation being 0.0003) for the other two adopted models. The magnetic spin density s_m_ is obtained through the sum of the magnetic orbital densities. Details on the evaluation of the electron spin density for the various considered wavefunction models are reported in Appendix A.

Topological analysis of the various scalar fields was performed through modified versions of the AIMPAC [43] and Multiwfn [44] program packages. The latter was employed for evaluating the spin density basins bounded by s = 0 isosurfaces and their properties. Details on the use and changes made to AIMPAC and Multiwfn packages are illustrated in Appendix A. The ball–and–stick pictures (Figure 3 and Figure 4 and Appendix A) were obtained through the code Diamond v3.21 [39], while VESTA *3* [45] and UCSF Chimera [40] were used for the 3D images. VESTA *3* was employed for the spin and magnetic orbitals densities isosurface representations (Appendix A) and UCSF Chimera for those of the basins bounded by s = 0 isosurfaces (Figure 5). 

## 5. Conclusions

In this work, the topology of the spin density distribution s(**r**) was systematically investigated for the first time and found to bear greater topological complexity than that of the electron density. Novel notions, such as spin graphs, spin basins and spin valence, and novel descriptors, such as the local/integral Spin Polarization Indices (SPI) or the basin average spin density, were introduced. It was then shown that two kinds of structures are associated with a spin–polarized molecule: the usual one, defined through the electron density gradient, and the *magnetic* structure, defined through the spin density gradient and composed in general by at least two independent spin graphs, related to spin density maxima and minima. After illustrating the general features of the spin density topology, the prototypical example of this function topology in the ^3^B_1_ water molecule was addressed in large detail, using spin density distributions of increasing accuracy. Local and nonlocal s(**r**) descriptors help to explain the ^3^B_1_ water real–space magnetic structure and to single out those features that are largely model dependent. 

Overall, we showed that the spin density topology can disclose a wealth of chemically and physically meaningful information. Most of the introduced spin density descriptors do not require the explicit knowledge of the system’s wavefunction, being therefore amenable to experimental investigation of the s(**r**) observable. Pioneering studies based on combined X–ray and neutron structure factors are already available; in the next future, it is foreseeable that a combination of accurate neutron detectors and more intense sources will disclose more and more reliable and precise experimental spin density distributions in crystalline materials. The topological toolbox here proposed might help in gaining insights into fascinating and not yet explored aspects of complex magnetic structures.

## Figures and Tables

**Figure 1 molecules-25-03537-f001:**
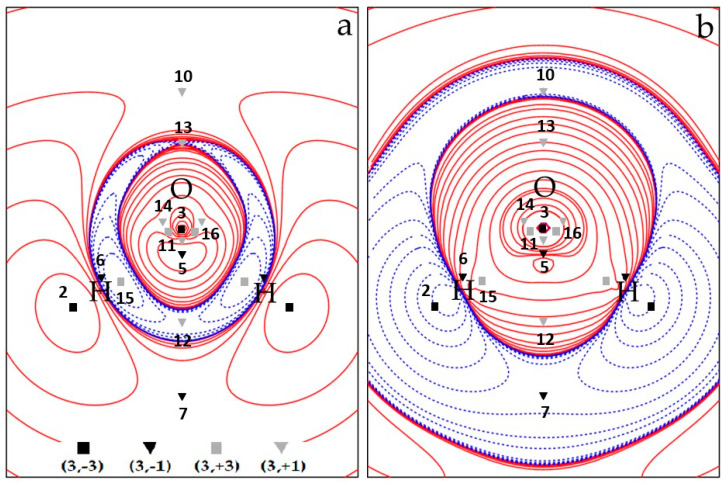
CAS contour plots in the σ_v_ plane of ^3^B_1_ H_2_O containing the three nuclei: (**a**) electron spin density s and (**b**) electron spin density Laplacian ∇^2^s. Contour maps are drawn in intervals of ± (2, 4, 8) × 10^−n^, 0 ≤ n ≤ 6 au (e^−^(bohr)^−3^ and e^−^(bohr)^−5^ for electron spin density and electron spin density Laplacian, respectively). Spin density CPs are shown on both maps, the unique ones labelled also by their identification number N (Table 2 and Table 3). CPs’ 6 locations are projected on this map (see text for more details). Solid red contour lines corresponds to s or ∇^2^s positive values, while the dashed blue contour lines correspond to negative s or ∇^2^s values. The solid squares mark (3, −3) CPs, the solid triangles (3, −1) CPs, gray triangles (3, +1) CPs and gray squares (3, +3) CPs. Note that, in this figure, in Figure 2 and in all figures of the Appendix A, the atomic nuclei are located in the close neighbourhood of the atomic labels.

**Figure 2 molecules-25-03537-f002:**
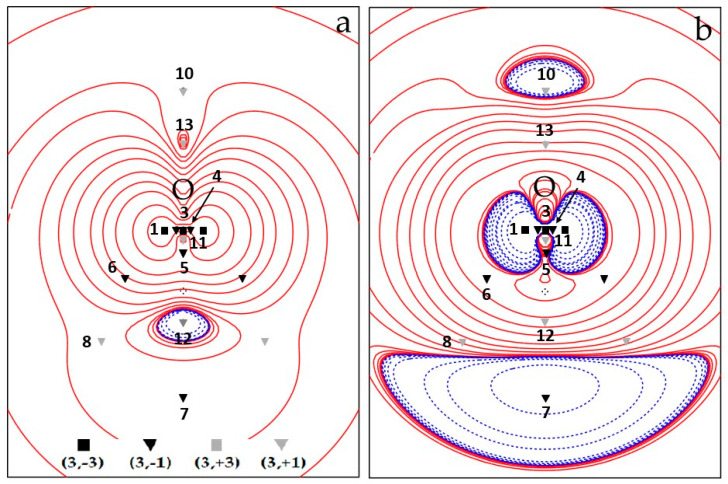
CAS contour plots in the σ_v’_ plane of ^3^B_1_ H_2_O containing the O nucleus: (**a**) electron spin density s and (**b**) electron spin density Laplacian ∇^2^s. For all other details, see the caption of Figure 1.

**Figure 3 molecules-25-03537-f003:**
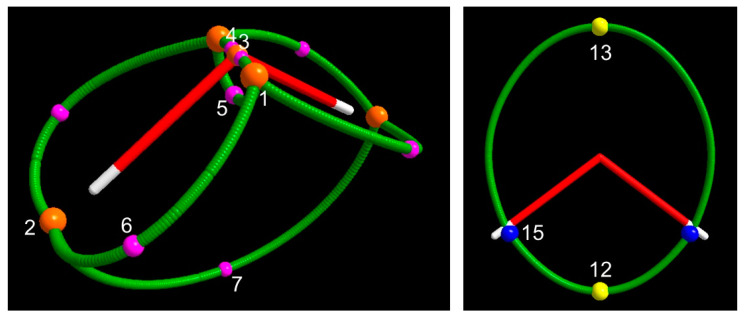
CAS spin density maxima joining paths: α–α, positive spin density regions (s_+_, left); β–β, negative spin density regions (s_−_, right). In both panels, the **∇**ρ bond paths are also shown (from H nucleus to the bond critical point (bcp) in white and from bcp to O nucleus in red). Critical points of s(**r**) are labelled as in Table 2 and Table 3 and portrayed as colored balls ((3, −3) s maxima, orange; (3, −1) s saddles, violet; (3, +3) s minima or –s maxima, blue; (3, +1) s ring or –s (3, −1) saddles, yellow). The α–α and β–β spin maxima joining paths are represented as thick green wires. The β–β spin density maxima joining paths all lie in the σ_v_ plane of ^3^B_1_ H_2_O, i.e., that defined by CPs 2, 2’ and 3 in the left panel. The pictures were obtained through the code Diamond v3.21 [39].

**Figure 4 molecules-25-03537-f004:**
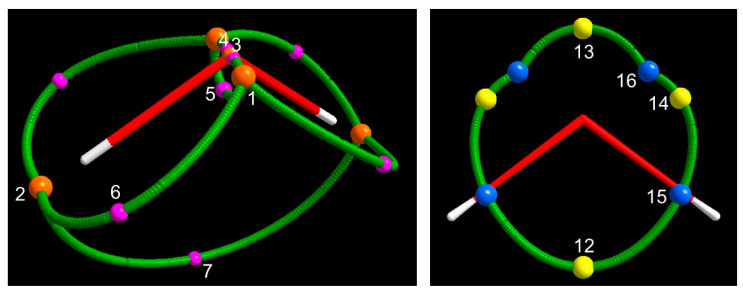
Unrestricted Hartree Fock **(**UHF) spin density maxima joining paths: α–α, positive spin density regions (s_+_, left); β–β, negative spin density regions (s_–_, right). In both panels, the **∇**ρ bond paths are also shown (from H nucleus to the bcp in white and from bcp to O nucleus in red). For all other details, see the caption of Figure 3.

**Figure 5 molecules-25-03537-f005:**
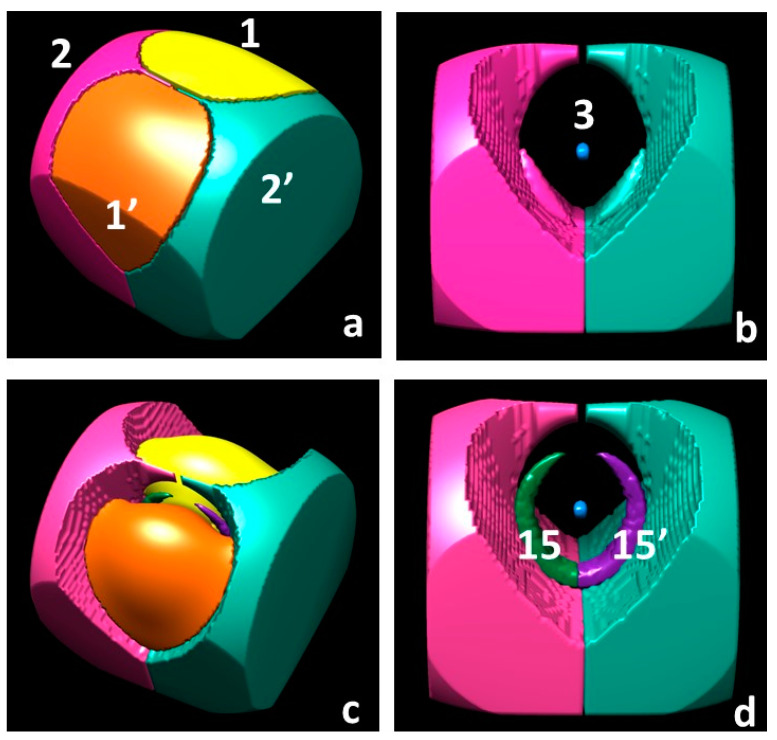
^3^B_1_ H_2_O CAS spin density basins. The basins are labelled by their enclosed (3, −3) spin maximum or, for the basin labelled 15, by its enclosed (3, +3) spin minimum (see Table 2 and Table 3). Each basin retains the same color in all illustrations. Panels **a** and **b** refer to basins bounded by local Zero–Flux Surfaces (ZFS) of **∇**s, while panels **c** and **d** to basins bounded by s = 0 isosurfaces (see text). Panel **b** (**d**) differs from panel **a** (**c**) for the removal of basins 1 and 1’, so disclosing the small and embedded basin 3 around the O nucleus. Basins are represented through their spin density isovalue surfaces ((bohr)^−3^): 0.0003 (basins 1,1’; **∇**s ZFS); 0.00009 (basins 1,1’; s = 0); 0.00031 (basins 2,2’); 0.0751 (basin 3); −0.000017 (basins 15, 15’). The drawings in the four panels were performed with UCSF Chimera, developed by the Resource for Biocomputing, Visualization, and Informatics at the University of California, San Francisco, with support from NIH P41–GM103311 [40].

**Table 1 molecules-25-03537-t001:** Classification of all possible non–degenerate spin density critical points CPs(*r*,*s*) in terms of their rank *r*, signature *s* and possible types, depending on −∇^2^ρ_α_ and −∇^2^ρ_β_ values.

CP(*r,s*)	−∇^2^s	Type	−∇^2^ρ_α_	−∇^2^ρ_β_	Constraint
(3, −3), max	> 0	1	> 0	> 0	−∇^2^ρ_α_ > −∇^2^ρ_β_
	> 0	2	> 0	< 0	None
	> 0	3	< 0	< 0	|−∇^2^ρ_α_| < |−∇^2^ρ_β_|

(3, −1), 1st order SP	> 0	1	> 0	> 0	−∇^2^ρ_α_ > −∇^2^ρ_β_
	> 0	2	> 0	< 0	None
	> 0	3	< 0	< 0	|−∇^2^ρ_α_| < |−∇^2^ρ_β_|
	< 0	4	> 0	> 0	−∇^2^ρ_α_ < −∇^2^ρ_β_
	< 0	5	< 0	> 0	None
	< 0	6	< 0	< 0	|−∇^2^ρ_α_| > |−∇^2^ρ_β_|
(3, +1), 2nd order SP	Same as for (3, −1)				
(3, +3), min	< 0	1	> 0	> 0	−∇^2^ρ_α_ < −∇^2^ρ_β_
	< 0	2	< 0	> 0	None
	< 0	3	< 0	< 0	|−∇^2^ρ_α_| > |−∇^2^ρ_β_|

**Table 2 molecules-25-03537-t002:** Selected properties ^1^ at the (3, −3) and (3, −1) critical points of the ^3^B_1_ H_2_O molecule for various wavefunction models.

		WFN	Type	s	s_mag_	ρ	∇ρ_α_ ≡ ∇ρ_β_	∇^2^s	∇^2^ρ_α_	∇^2^ρ_β_	SPI(r_c_)
**Maxima (3, −3)**											
1	2	ROHF	*3*	0.603	0.603	1.938	8.715	−26.978	81.209	108.186	1.269
	2	UHF	*3*	0.613	0.607	1.816	7.714	−25.944	71.334	97.278	1.346
	2	CAS	*3*	0.617	0.615	1.820	7.681	−25.670	71.105	96.775	1.351

2	2	ROHF	*1*	0.018	0.018	0.263	0.341	−0.697	−5.517	−4.821	0.765
	2	UHF	*2*	0.015	0.016	0.090	0.123	−0.119	−0.091	0.028	0.923
	2	CAS	*3*	0.012	0.014	0.072	0.096	−0.082	0.003	0.086	0.936

3	1	ROHF	*1*	0.683	0.683	295.534	557.768	−5.6 × 10^3^	−1.2 × 10^6^	−1.2 × 10^6^	0.670
	1	UHF	*1*	0.840	0.706	295.432	587.366	−6.7 × 10^3^	−1.2 × 10^6^	−1.2 × 10^6^	0.670
	1	CAS	*1*	1.262	0.725	295.814	408.328	−10.1 × 10^3^	−1.2 × 10^6^	−1.2 × 10^6^	0.672
**Saddles (3, −1)**											
4	2	ROHF	*1*	0.367	0.367	82.250	648.694	−45.534	−4.85 × 10^3^	−4.80 × 10^3^	0.673
	2	UHF	*1*	0.415	0.380	64.569	503.389	−47.064	−1.91 × 10^3^	−1.86 × 10^3^	0.675
	2	CAS	*1*	0.474	0.417	41.885	323.092	−45.172	−66.122	−20.950	0.682

5	1	ROHF	*6*	0.094	0.094	1.265	1.691	2.051	8.785	6.734	0.773
	1	UHF	*6*	0.130	0.100	1.308	1.921	1.206	11.013	9.807	0.813
	1	CAS	*6*	0.147	0.102	1.327	2.032	0.850	11.728	10.877	0.833

6	4	ROHF	*6*	0.006	0.006	0.085	0.098	0.023	0.099	0.077	0.744
	4	UHF	*6*	0.005	0.006	0.048	0.057	0.013	0.086	0.073	0.819
	4	CAS	*6*	0.004	0.005	0.039	0.047	0.010	0.076	0.066	0.826

7	1	ROHF	*3*	0.003	0.003	0.006	0.003	−0.002	0.003	0.006	2.455
	1	UHF	*3*	0.003	0.004	0.006	0.003	−0.003	0.003	0.005	2.575
	1	CAS	*3*	0.003	0.003	0.006	0.003	−0.003	0.002	0.005	2.581

^1^ Dimensioned property in au (e^−^(bohr)^−3^ for electron and electron spin densities; e^−^(bohr)^−4^ for electron and electron spin density gradients; e^−^(bohr)^−5^ for electron and electron spin density Laplacians). For each CP, the CPs identification number (N), multiplicity (M) and type (see Table 1) is reported along with the values of several properties at the CP: the electron spin density s and its magnetic component s_mag_, the electron density ρ, the gradient of the α– and β–electron density ∇ρ_α_(≡ ∇ρ_β_), the Laplacian of the spin density ∇^2^s, its α and β–components ∇^2^ρ_α_ and ∇^2^ρ_β_ and the Spin Polarization Index at the CP, SPI(**r**_c_) (see text for definition).

**Table 3 molecules-25-03537-t003:** Selected properties ^1^ at the (3, +1) and (3, +3) critical points of the ^3^B_1_ H_2_O molecule for various wavefunction models.

N	M	WFN	Type	s	s_mag_	ρ	∇ρ_α_ ≡ ∇ρ_β_	∇^2^s	∇^2^ρ_α_	∇^2^ρ_β_	SPI(r_c_)
8	2	ROHF	*6*	0.003	0.003	0.009	0.007	0.004	0.018	0.014	1.252
	2	UHF	*6*	0.003	0.003	0.008	0.006	0.003	0.015	0.012	1.315
	2	CAS	*6*	0.002	0.003	0.007	0.006	0.003	0.014	0.011	1.350

9	2	ROHF	*6*	0.001	0.001	0.039	0.055	0.007	0.108	0.101	0.694

10	1	UHF	*3*	0.000	0.000	0.0051	0.0058	−0.0003	0.0082	0.0085	0.754
	1	CAS	*3*	0.000	0.000	0.0056	0.0061	−0.0002	0.0083	0.0085	0.750

11	1	UHF	*6*	0.011	0.000	25.929	195.627	52.020	637.129	585.109	0.667
	1	CAS	*6*	0.030	0.001	20.991	156.382	49.062	692.826	643.765	0.669

12	1	UHF	*6*	−0.001	0.000	0.062	0.077	0.055	0.191	0.136	0.644
	1	CAS	*6*	−0.002	0.000	0.063	0.078	0.059	0.194	0.136	0.631

13	1	UHF	*6*	−0.001	0.001	0.297	0.409	0.176	0.321	0.145	0.661
	1	CAS	*6*	−0.000	0.001	0.068	0.095	0.021	0.159	0.138	0.667

14	2	UHF	*6*	−0.002	0.000	0.225	0.289	0.166	0.390	0.224	0.654
	2	CAS	*6*	0.023	0.000	1.517	3.758	7.252	38.928	31.677	0.687
15	2	UHF	*1*	−0.007	0.000	0.295	0.063	0.261	−0.577	−0.837	0.638
	2	CAS	*1*	−0.008	0.001	0.289	0.053	0.260	−0.696	−0.956	0.631

16	2	UHF	*2*	−0.002	0.000	0.390	0.508	0.380	0.289	−0.091	0.658
	2	CAS	*3*	0.020	0.000	6.369	42.376	23.418	346.437	323.019	0.671

^1^ Dimensioned property in au (e^−^(bohr)^−3^ for electron and electron spin densities; e^−^(bohr)^−4^ for electron and electron spin density gradients; e^−^(bohr)^−5^ for electron and electron spin density Laplacians). For the description of listed properties, see footnote 1 of Table 2.

**Table 4 molecules-25-03537-t004:** Basin Ω Selected Properties of the ^3^B_1_ H2O molecule for various Ω basin types and wavefunction (WFN) models ^1^.

Ω	M	WFN	N_Ω_	SP_Ω_	SP_mag,Ω_	SPI_Ω_	V1_Ω_	V2_Ω_	${\bar{s}}_{\Omega }$
**∇ρ ZFS boundaries**									
O	1	ROHF	8.867	1.409	=	0.919	180.6	127.4	0.008
	1	UHF	8.856	1.417	1.409	0.921	179.2	126.8	0.008
	1	CAS	8.743	1.406	1.387	0.922	173.3	122.5	0.008
H	2	ROHF	0.565	0.294	=	2.113	71.8	39.8	0.004
	2	UHF	0.571	0.290	0.295	2.046	72.2	40.3	0.004
	2	CAS	0.628	0.296	0.306	1.856	76.9	42.6	0.004
**∇s ZFS boundaries**									
Ω_1_	2	ROHF	3.469	0.587	=	0.938	53.9	42.0	0.011
	2	UHF	3.740	0.597	0.591	0.920	52.1	40.9	0.011
	2	CAS	3.620	0.604	0.595	0.934	52.4	40.9	0.012
Ω_2_	2	ROHF	0.901	0.409	=	1.773	108.3	61.4	0.004
	2	UHF	0.907	0.402	0.407	1.727	109.8	62.7	0.004
	2	CAS	0.913	0.394	0.403	1.676	111.2	63.0	0.004
Ω_3_	1	ROHF	1.231	0.006	=	0.673	0.04	0.04	0.174
	1	UHF	0.699	0.003	0.002	0.672	0.01	0.01	0.196
	1	CAS	0.935	0.005	0.002	0.674	0.02	0.02	0.231
**s = 0 isovalue surface boundaries**									
Ω_1_	2	UHF	3.574	0.598	0.587	0.935	50.9	39.7	0.012
	2	CAS	3.509	0.606	0.589	0.945	51.6	40.2	0.012
Ω_2_	2	UHF	0.836	0.403	0.409	1.906	109.4	62.3	0.004
	2	CAS	0.821	0.395	0.407	1.901	110.6	62.3	0.004
Ω_3_	1	UHF	0.640	0.002	0.002	0.671	0.01	0.01	0.210
	1	CAS	0.916	0.005	0.004	0.674	0.02	0.02	0.237
Ω_15_	2	UHF	0.173	−0.002	0.002	0.653	1.02	1.02	−0.002
	2	CAS	0.212	−0.003	0.003	0.650	1.45	1.45	−0.002
_Ω16_	2	UHF	0.097	−0.000	0.001	0.660	0.68	0.68	−0.001

^1^ Dimensioned property in au (e^−^ for electron or spin populations and (bohr)^3^ for volumes). Properties for basins bounded by local Zero–Flux Surfaces (ZFS) of the gradient of the electron or of the electron spin densities, as well as for basins bounded by s = 0 isovalue surfaces are reported. The last two kind of basins are denoted as Ω_x_, where x denotes their (3, −3) s maxima (Table 2) or, for basins of s_−_ regions, their associated (3, +3) CP (Table 3). M, N_Ω_, SP_Ω_, SP_mag,Ω_ and SPI_Ω_ are, respectively, the multiplicity, the electron population, the spin population, the spin population magnetic component and the Spin Polarization Index of the basin Ω (see text). The basin volumes V1_Ω_ and V2_Ω_ are defined, for all basin types, in terms of the electron density values (V1, ρ ≥ 0.001 e^−^(bohr)^−3^**;** V2, ρ ≥ 0.002 e^−^ (bohr)^−3^). ${\bar{s}}_{\Omega }=SP_{\Omega }/V1_{\Omega }$ is the basin Ω average spin density.

**Table 5 molecules-25-03537-t005:** ^3^B_1_ H_2_O molecule: Source Function percentage contributions (SF%) to the electron spin density CPs for various Ω Basin types ^1^.

Ω	SF%_CP1_	SF%_CP2_	SF%_CP3_	SF%_CP4_	SF%_CP5_	SF%_CP6_	SF%_CP7_
**∇s ZFS boundaries**							
Ω_1_	94.53	−0.12	8.23	61.82	50.40	−0.08	1.30
Ω_1’_	3.91	−0.12	8.23	10.72	50.40	36.88	1.30

_Ω2_	0.05	96.50	0.02	0.06	0.14	51.63	47.20
Ω_2’_	0.05	2.72	0.02	0.06	0.14	7.81	47.20

Ω_3_	1.46	1.02	83.49	27.35	−1.08	3.77	3.00

**∇ρ ZFS boundaries**							
O	99.29	−33.98	99.62	99.00	96.45	−62.50	−75.30

H	0.39	121.21	0.19	0.50	1.77	123.05	87.60
H’	0.39	12.77	0.19	0.50	1.77	39.54	87.60

^1^ Properties for basins bounded by local Zero–Flux Surfaces (ZFS) of the gradient of the electron spin density or of the electron density are listed in the Table. Basins are labelled as in Table 4. The source function percentage contribution from each basin Ω is listed in the corresponding row and labelled as SF%_CPX_, where x denotes the specific (3, −3) or (3, −1) spin density CP it refers to (Table 2). When a CP is differently distant from the attractor of a unique basin Ω_y_ having a multiplicity M > 1 (Table 4), the farther basin is the primed one (Ω_y_’).

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
