# Peer review of "Spin Density Topology"

_molecules, 2020, doi:10.3390/molecules25153537_

Round 1

Reviewer 1 Report

With "Spin Density Topology", the authors present a study of qualitative features of the spin density, i.e., the difference of the up- und down-spin electron density. After some general discussion, the authors exemplify their introduced quantities with the 3B1 state of the water molecule.

Personally, I'm not a big fan of chemical topology and I have never understood how it can be useful. However, just because I don't like it does not mean that it shouldn't be done! This comment is just to make it clear that I do not plan to judge the relevance of the article.

In any case, the article presents solid scientific work in a suitable way. The electron density (and thus the spin density) is a reduced quantity and its features are much more subtle than those of the many-electron wavefunction; often, the electron density of ground and excited states is barely distinguishable. Studying the number and location of extrema of the electron density can be a way to characterize the electron density, and thus also the spin density. As far as I can see, this was comprehensively done and I did not detect any methodological errors or problems. The discussion of the water molecule illustrates the introduced quantities but it also illustrates that those are rather sensitive to the electronic structure method, which is certainly not a good sign.

The article is also one of the few articles that I reviewed where I think that it can be essentially published "as is". Personally, I would have mixed introduction of the quantities and exemplifying them with the water molecule to make the article easier to read, but it is also OK as it currently is. I like that you give some technical hints in the Supplement. Nevertheless, I would recommend that you read the article carefully again because there are quite a few grammatical errors or problems. Here are some of the ones that I found:

L 13: "and its being" -> "and it being"
L 19: "the different 18 ways by which they physically take place" -> "the 18 different ways in which they can be realized"
L 44: "enabled" -> "possible"
L 113: "one with the other" -> "each other"
L 144: "was proved" -> "was proven" or, better, "has been proven"
L 146: "M. Rehier" -> "M. Reiher"
L 265/277 (& more?): choose how to write "non-degenerate"
L 297: "infinite" -> "infinity"
L 397/731: This triple fraction is really not necessary, just multiply by N_beta/N_alpha
L 418: "are so only"
L 423: "plane"
L 433 (& more): "au" is not a unit; quantities in atomic units have a unit, please get them from the Wikipedia article or from any source you like
L 609: your abbreviations "4MR" and "3MR" already include the "ring", so it should not be written anymore
L 727: "basins" -> "basin"
L 875: Even though some popular program packages call it "geometry optimization", the geometry is not optimized...it is still Euclidean, I hope! The "nuclear structure" or "nuclear configuration" may be optimized.

Supplement:
L 75 & more: "in input" -> "as input"

Author Response

We gratefully thank the reviewer for his/her very careful reading of our manuscript and for his/her globally positive judgement.

We have revised our text and SI following all suggestions kindly provided by the referee. Changes are highlighted in yellow in the revised manuscript and SI.

Here follows a point-by-point response to the reviewer's comments:

          L 13: "and its being" -> "and it being"

Reply: corrected as requested

           L 19: "the different 18 ways by which they physically take place" -> "the 18 different ways in which they can be realized"

Reply: corrected as requested

           L 44: "enabled" -> "possible"

Reply: corrected as requested

           L 113: "one with the other" -> "each other"

Reply: corrected as requested

           L 144: "was proved" -> "was proven" or, better, "has been proven"

Reply: corrected as requested ("has been proven")

           L 146: "M. Rehier" -> "M. Reiher"

Reply: corrected as requested

           L 265/277 (& more?): choose how to write "non-degenerate"

Reply: in all occurrences throughout the text we have now adopted the  "non-degenerate"  writing

           L 297: "infinite" -> "infinity"

Reply: corrected as requested

           L 397/731: This triple fraction is really not necessary, just multiply by N_beta/N_alpha

Reply: The expression has been changed accordingly to the reviewer suggestion at both lines  (referee line numbering)

            L 418: "are so only"

Reply: We have decided to keep our expression : "There are so only" because it seems to us more correct. Also Referee 4, which has asked us to modify several expressions, was fully satisfied by our original expression in this specific case.

L 423: "plane"

Reply: We have corrected the first occurrence of "plane" in this line to "planes"

L 433 (& more): "au" is not a unit; quantities in atomic units have a unit, please get them from the Wikipedia article or from any source you like

Reply: We have inserted the specific  units for each quantity:

a) in the caption of Table 2:

Dimensioned property in au [e-(bohr)-3 for electron and electron spin densities; e-(bohr)-4 for electron and electron spin density gradients; e-(bohr)-5 for electron and electron spin density Laplacians].

b) In the caption of Figure 1:

Contour maps are drawn in intervals of ± (2,4,8)∙ , 0 ≤ n ≤ 6 au [e-(bohr)-3 and e-(bohr)-5 for electron spin density and electron spin density Laplacian, respectively].

c) at line 557 (our line numbering)  [> 0.1 e-(bohr)-3 ]. 

d) in the caption of Table 3: Dimensioned property in au [e-(bohr)-3 for electron and electron spin densities; e-(bohr)-4 for electron and electron spin density gradients; e-(bohr)-5 for electron and electron spin density Laplacians]. 

e) at line 666 (our line numbering) : [0.002-0.007 e-(bohr)-3

f) at line 671 (our line numbering) : being -1×10-5 e-(bohr)-3]. 

g) at line 678 (our line numbering) : [0.020 and 0.023 e-(bohr)-3, respectively]. 

h) in the caption of Table 4: 1) Dimensioned property in au [e- for electron or spin populations and (bohr)3 for volumes] and 2) in terms of the electron density values [V1, r ³ 0.001 e-(bohr)-3]; V2, r ³ 0.002 e-(bohr)-3].

i) at line 775 (our line numbering) : "or exceeds 0.001 or 0.002 e-(bohr)-3

j)  at line 804 (our line numbering) : "[V1 = 0.02 (bohr)3, CAS]"

l) at line 829 (our line numbering) :[2.9 and 3.4 (bohr)3

m) at line 830 (our line numbering): [-0.002 (bohr)-3]

n) caption Figure 5 line 840 (our line numbering): [(bohr)-3]: 

o) caption Table S3 : Dimensioned property in au [e-(bohr)-3 for electron and electron spin densities; e-(bohr)-4 for electron and electron spin density gradients; e-(bohr)-5 for electron and electron spin density Laplacians]. 

p) captionTable S4:

1 Dimensioned property in au [e-(bohr)-3 for electron and electron spin densities; e-(bohr)-4 for electron and electron spin density gradients; e-(bohr)-5 for electron and electron spin density Laplacians]. For the description of listed properties see footnote 1 of Table S3

q) caption Figure S1, line 144: namely s = 0.0015 e-(bohr)-3 (red) and s = -2×10-8 e-(bohr)-3 (dark red). 

r) caption Figure S2:  Contour maps are drawn in intervals of ± (2,4,8)∙, 0 ≤ n ≤ 6 au [e-(bohr)-3 and e-(bohr)-5 for electron spin density and electron spin density Laplacian, respectively].

s) Caption Figure S6 : Contour maps are drawn in intervals of ± (2,4,8)∙, 0 ≤ n ≤ 6 au [e-(bohr)-3 and e-(bohr)-5 for electron spin density and electron spin density Laplacian, respectively]. 

             L 609: your abbreviations "4MR" and "3MR" already include the "ring", so it should not be written anymore

Reply: we have followed the advise of the Referee everywhere in the text, when necessary.

              L 727: "basins" -> "basin"

Reply: corrected as requested

               L 875: Even though some popular program packages call it "geometry optimization", the geometry is not optimized...it is still Euclidean, I hope! The "nuclear structure" or "nuclear configuration" may be optimized.

Reply: corrected as suggested

Supplement:
                L 75 & more: "in input" -> "as input"

Reply: cofrrecetd everywhere as requested

Reviewer 2 Report

This is an excellent piece of work, introducing a new means of analysing the electronic structure of molecules. It is deeply thought out and thoroughly presented, and will be of great interest to many theoretical and experimental chemists. It sets out new ways to examine the spin density within open-shell molecules, which is important for many phenomena.

I am pleased to recommend publication in current form.

Author Response

We gratefully thank the referee for his/her careful reading of our manuscript and for his/her very positive judgment.

English language and style are fine/minor spell check required : changes (highlighted in yellow) to the manuscript were made according to the suggestions provided by Referees 1 and 4.

Reviewer 3 Report

Topological analysis of spin density as described in this interesting paper gives a new view into chemistry. I love to see this paper published.   

Author Response

We gratefully thank the referee for his/her careful reading of our manuscript and for his/her very positive judgment.

Reviewer 4 Report

This paper on "Spin Density Topology" by G. Bruno et al. is a fine piece of work on the topology of a scalar function with great importance in open-shell systems: the "spin density" (or, maybe better, the difference of the spin-dependent densities). I consider this work being very well suited for publication in the journal "Molecules". Only a few minor points, in addition to the correction of typos (see attached PDF file), require modification or revision and hence should be addressed by the authors when they prepare a final version for publication:

(1) Strictly spoken, there is no such thing as "the water triplet molecule" (with definite article) or a "water molecule in its triplet state". There are several (or many) possible excited triplet states of the water molecule. The case under study in this work should be named completely(!) and as early as possible. The statement of an electron configuration (at ROHF at least) would be very helpful, even though it is only approximate information (see spin-contamination annihilated UHF and CASSCF levels). In this context, it must not be forgotten that a complete description of a quantum mechanical state requires specification of all eigenvalues for the complete set of commuting operators (CSCO, see, e.g., Dirac, The Principles of Quantum Mechanics, 1930). For the present study, this means that specification of quantum number M_S is required (M_S=+1, if α is majority spin). Much of the analyses presented here is applicable to the case M_S=-1 (with β as the majority spin) or can be transferred to that case with ease. But what about M_S=0 (which has to have exactly the same total energy as the two |M_S|=1 cases, as long as a magnetic field is absent)?? So, dear authors, please specify M_S, for completeness and for clearliness, and do so early! In view of intended application to open-shell transition metal compounds, a comment on applicability of the new tools to non-maximal |M_S| states would be very much appreciated too!

(2) The term "interaction" has been used in the discussion of gradient paths, see e.g. sec. 2.1.3. I consider this choice of terminology as a very unfortunate (potentially even disastrous) choice, for the following two reasons: (i) the only "physical" interactions in this system are of electromagnetic nature (or, simplistically, as exploited here, just electrostatic [Coulomb interactions]) --- and as long as you do not first perform a decent quantum-chemical calculation, based on and accounting for these "physical" interactions, for the system under study (as we fortunately nowadays know how to do) you get absolutely nothing into your hands to which your topological analysis can be applied; (ii) this work discusses topological properties of a scalar field (a very important scalar field: the difference density or "spin density" [one may expect a vector field from this latter term]) --- consequently the (available) terminology from that branch of science should be used and is sufficient (maybe not for the average chemist, who is too little trained in topology). No need for the invention of potentially misleading "new" terminology. Otherwise, if the term "interaction" and all others closely related to it remain in this work, then it can happen that in, say, two or five years from now people give talks on conferences and claim that they "really believe" in "interaction between spins along spin maxima/minima interaction lines" ... poor physics.

Author Response

We gratefully thank the referee for his/her very careful reading of our manuscript as well as for his/her overall positive judgement. We greatly thank   him/her also for the two extremely relevant concerns he/she raised.

Point-by-point response to the rviewer's comments:

              a) Only a few minor points, in addition to the correction of typos (see attached PDF file), require modification or revision..

Reply: The required modification have all been addressed, following the suggestions provided by the referee in his/her annotated pdf file. Changes are highlighted in yellow in the revised manuscript.

Only a few minor points, in addition to the correction of typos (see attached PDF file), require modification or revision and hence should be addressed by the authors when they prepare a final version for publication:

(1) Minor point :  Strictly spoken, there is no such thing as "the water triplet molecule" (with definite article) or a "water molecule in its triplet state". There are several (or many) possible excited triplet states of the water molecule. The case under study in this work should be named completely(!) and as early as possible. The statement of an electron configuration (at ROHF at least) would be very helpful, even though it is only approximate information (see spin-contamination annihilated UHF and CASSCF levels). In this context, it must not be forgotten that a complete description of a quantum mechanical state requires specification of all eigenvalues for the complete set of commuting operators (CSCO, see, e.g., Dirac, The Principles of Quantum Mechanics, 1930). For the present study, this means that specification of quantum number M_S is required (M_S=+1, if α is majority spin). Much of the analyses presented here is applicable to the case M_S=-1 (with β as the majority spin) or can be transferred to that case with ease. But what about M_S=0 (which has to have exactly the same total energy as the two |M_S|=1 cases, as long as a magnetic field is absent)?? So, dear authors, please specify M_S, for completeness and for clearliness, and do so early! In view of intended application to open-shell transition metal compounds, a comment on applicability of the new tools to non-maximal |M_S| states would be very much appreciated too!

Reply: We fully agree with the reviewer's comment.  The case under study in our work (water molecule in the 3B1 triplet state) has now been named completely, right from the beginning (in the abstract and in all the following parts of the manuscript whenever it was used the imprecise terminology ["the water triplet molecule" (with definite article) or a "water molecule in its triplet state"]

(2) Minor point :  The term "interaction" has been used in the discussion of gradient paths, see e.g. sec. 2.1.3. I consider this choice of terminology as a very unfortunate (potentially even disastrous) choice, for the following two reasons: (i) the only "physical" interactions in this system are of electromagnetic nature (or, simplistically, as exploited here, just electrostatic [Coulomb interactions]) --- and as long as you do not first perform a decent quantum-chemical calculation, based on and accounting for these "physical" interactions, for the system under study (as we fortunately nowadays know how to do) you get absolutely nothing into your hands to which your topological analysis can be applied; (ii) this work discusses topological properties of a scalar field (a very important scalar field: the difference density or "spin density" [one may expect a vector field from this latter term]) --- consequently the (available) terminology from that branch of science should be used and is sufficient (maybe not for the average chemist, who is too little trained in topology). No need for the invention of potentially misleading "new" terminology. Otherwise, if the term "interaction" and all others closely related to it remain in this work, then it can happen that in, say, two or five years from now people give talks on conferences and claim that they "really believe" in "interaction between spins along spin maxima/minima interaction lines" ... poor physics.

Reply: We gratefully thank the referee for this very important warning. We have therefore replaced, everywhere in the text and in the SI,  "spin maxima interaction lines" with "spin maxima joining paths". These paths are indeed the union of the two unique spin density gradient paths, originating at a (3,-1) CP and ending up at their own (3,-3) CP attractor. As such they turn out to be paths joining two spin density maxima, in formal analogy with the "bond path" of the electron density topology. We think to have so avoided any possible misunderstanding related to "spin-spin interactions" which are clearly not taken into account by our electrostatic Hamiltonian.